# Towards a General Time Series Forecasting Model with Unified Representation and Adaptive Transfer

**Yihang Wang** [* 1] **Yuying Qiu** [* 1] **Peng Chen** [1] **Kai Zhao** [2] **Yang Shu** [1] **Zhongwen Rao** [3] **Lujia Pan** [3] **Bin Yang** [1] **Chenjuan Guo** [1]

## Abstract

With the growing availability of multi-domain time series data, there is an increasing demand for general forecasting models pre-trained on multi-source datasets to support diverse downstream prediction scenarios. Existing time series foundation models primarily focus on scaling up pre-training datasets and model sizes to enhance generalization performance. In this paper, we take a different approach by addressing two critical aspects of general forecasting models: (1) how to derive unified representations from heterogeneous multi-domain time series data, and (2) how to effectively capture domain-specific features to enable adaptive transfer across various downstream scenarios. To address the first aspect, we propose Decomposed Frequency Learning as the pre-training task, which leverages frequency-based masking and reconstruction to decompose coupled semantic information in time series, resulting in unified representations across domains. For the second aspect, we introduce the Time Series Register, which captures domain-specific representations during pre-training and enhances adaptive transferability to downstream tasks. Our model achieves the state-of-the-art forecasting performance on seven real-world benchmarks, demonstrating remarkable few-shot and zero-shot capabilities.

## 1. Introduction

Time series forecasting plays a crucial role in various domains, including energy, smart transportation, weather, and economics (Qiu et al., 2024; Wu et al., 2025a; Qiu et al., 2025b). However, training deep learning models for each specific dataset is resource-intensive and requires tailored parameter tuning. This approach often suffers from limited prediction accuracy due to data scarcity (Liu et al., 2024; Wu et al., 2021b; 2024; 2025c). A promising solution is to pre-train a general model on diverse time series datasets, which can then be fine-tuned with minimal data for different downstream scenarios or even used directly without fine-tuning. Following this idea, foundation models for time series forecasting have gained significant attention. Recent efforts have focused on scaling up pre-training datasets and model sizes to enhance generalization performance (Woo et al., 2024; Goswami et al., 2024; Ansari et al., 2024). However, excessive scaling introduces high computational costs during training and inference, undermining the practicality of general models, particularly in resource-constrained settings. Beyond scaling, the design of general time series forecasting models can also be approached through pre-training tasks and downstream transfer adaptation. From these two perspectives, we identify the following challenges.

**Obtaining a unified representation from time series data across various domains is challenging**. Time series from each domain involve complex temporal patterns, composed of multiple frequency components combined with each other (Zhou et al., 2022; Wu et al., 2025b), which is frequency superposition. As shown in Figure 1(a), different frequency components contain distinct semantic information. For example, low and high-frequency components represent long-term trends and rapid variations, respectively (Zhang et al., 2022). Furthermore, different datasets exhibit diverse frequency distributions, and the significance of low-frequency and high-frequency components for time series modeling varies across domains(Zhang et al., 2024). As a result, large-scale time series data from different domains introduce even more complex temporal patterns and frequency diversity. Existing pre-training frameworks (Dong et al., 2024; Nie et al., 2022; Lee et al., 2023), such as masked modeling and contrastive learning, were proposed to learn a unified representation from time domain. However, these methods overlook the frequency diversity and complexity exhibited in heterogeneous time series that come

---

[*]Equal contribution [1]East China Normal University, Shanghai, China [2]Aalborg University, Aalborg, Denmark [3]Huawei Noah's Ark Lab, Shenzhen, China. Correspondence to: Zhongwen Rao <raozhongwen@huawei.com>, Chenjuan Guo <cjguo@dase.ecnu.edu.cn>.

*Proceedings of the $42^{nd}$ International Conference on Machine Learning*, Vancouver, Canada. PMLR 267, 2025. Copyright 2025 by the author(s).

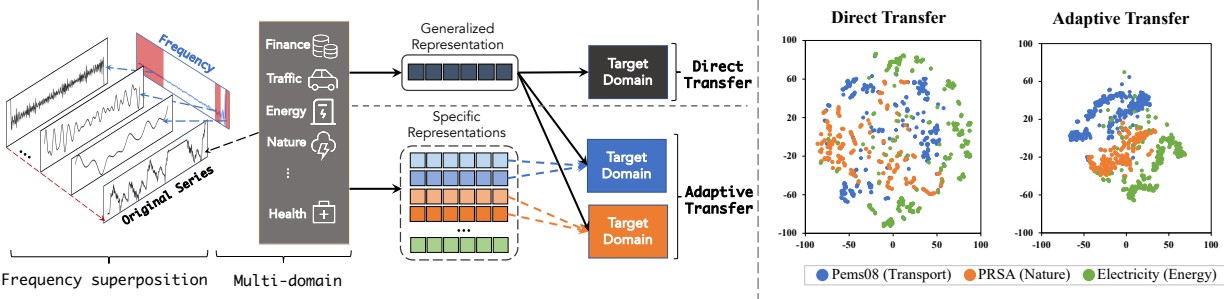

**(a) the illustration of general time series forecasting**  **(b) t-SNE visualization**

*Figure 1.* **(a)** Pre-training on multi-domain datasets that exhibit combined frequency. Existing general time series forecasting models only extract generalized representations for direct transfer to various downstream target domains. We propose to learn generalized and specific representations during pre-training, and adaptively transfer them to each target domain. **(b)** The t-SNE visualization of the hidden representations after direct transfer and adaptive transfer: In direct transfer, representations of different domains are mixed, but in adaptive transfer, they show a clear clustering pattern. The detailed experiment setting is in the Appendix A.1.3

from various domains, making it difficult to capture intricate patterns, thus limiting their generalization capabilities.

**Adaptive transferring information from multi-domain time series to specific downstream scenarios presents a challenge**. Multi-source time series data originate from various domains (Woo et al., 2024), whose data exhibit domain-specific information (Liu et al., 2024; Miao et al., 2024; Zhao et al., 2023; Guo et al., 2014). Information from the same or similar domain as the target domain is useful for improving the model's effectiveness in the target task (Chen et al., 2023a). However, as shown in Figure 1(a), existing time series pre-training frameworks (Woo et al., 2024; Liu et al., 2024; Zhou et al., 2024) primarily focus on learning generalized time series representations during pre-training while overlooking domain-specific representations, called *direct transfer*. While generalized representations are essential, directly transferring them to specific downstream tasks without incorporating domain-specific information leaves room for improvement. Therefore, it is necessary to learn domain-specific information during pre-training and adaptively transfer the specific representations to target domain, called *adaptive transfer*. Realizing adaptive transfer poses two difficulties: 1) capturing domain-specific information in pre-training. 2) adaptive use of domain-specific information in various downstream tasks.

To address these challenges, we propose a register assisted general time series forecasting model with decomposed frequency learning, namely **ROSE**. **First**, we propose Decomposed Frequency Learning that learns generalized representations to solve the issue with coupled semantic information. We decompose individual time series using the Fourier transform with a novel frequency-based masking method, and then convert it back to the time domain to obtain decoupled time series for reconstruction. It makes complex temporal patterns disentangled, thus benefiting the model to learn gen-

eralized representations. **Second**, we introduce Time Series Register (TS-Register) to learn domain-specific information in multi-domain data. By setting up a register, we generate register tokens to learn each domain-specific information during pre-training. In a downstream scenario, the model adaptively selects Top-K vectors from the register that are close to the target domain of interest. During fine-tuning, we adjust the selected register tokens with a novel learnable low-rank matrix, which complements target-specific information to perform more flexible adaptive transfer. As shown in Figure 1(b), adaptive transfer successfully utilizes domain-specific information in multi-domain time series, which contributes to the model's performance in target tasks. The contributions are summarized as follows:

- We propose ROSE, a novel light weight general time series forecasting model using multi-domain datasets for pre-training and improving downstream fine-tuning performance and efficiency.

- We propose a novel Decomposed Frequency Learning that employs multi-frequency masking to learn complex general temporal patterns from multi-domain data, empowering the model's generalization capability.

- We propose a novel TS-Register to capture domain-specific information in pre-training and enable adaptive transfer of target-oriented specific information for downstream tasks.

- Our experiments with 7 real-world benchmarks demonstrate that ROSE achieves state-of-the-art performance in full-shot setting and achieves competitive or superior results in few-shot setting, along with impressive transferability in zero-shot setting.

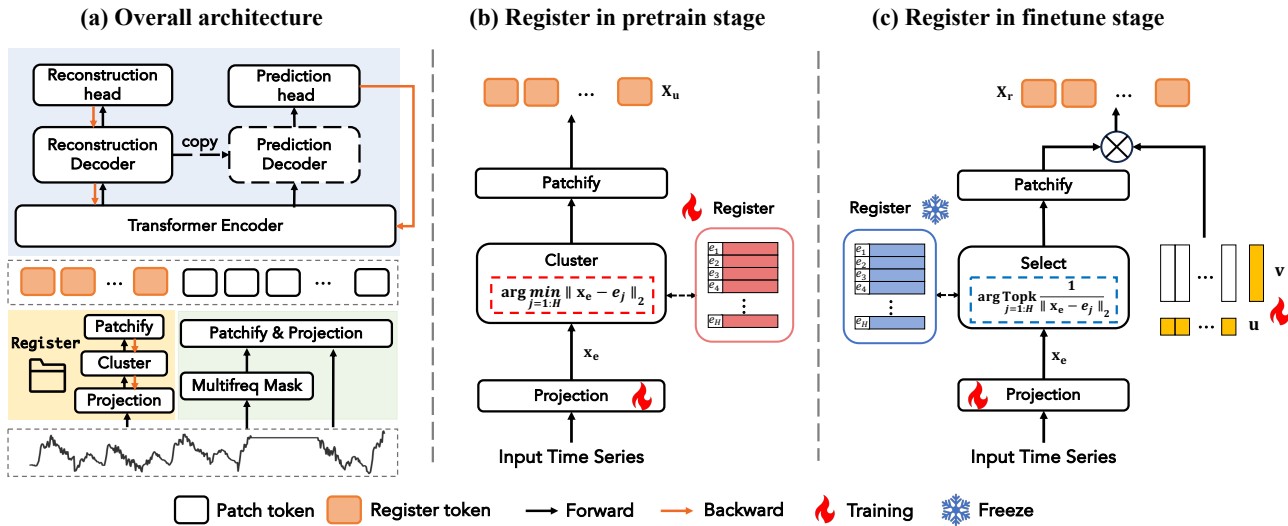

*Figure 2.* The model architecture of ROSE.

## 2. Related Work

### 2.1. Traditional Time Series Forecasting

The statistical time series forecasting models like ARIMA (Box & Jenkins, 1968), despite their theoretical support, are limited in modeling nonlinearity. With the rise of deep learning, many RNN-based models (Cirstea et al., 2019; Wen et al., 2017; Salinas et al., 2020) have been proposed, modeling the sequential data with an autoregressive process. CNN-based models (Luo & Wang, 2024; Liu et al., 2022b) have also received widespread attention due to their ability to capture local features. MICN (Wang et al., 2022) utilizes TCN to capture both local and global features, while TimesNet (Wu et al., 2022) focuses on modeling 2D temporal variations. However, both RNNs and CNNs struggle to capture long-term dependencies. Transformer-based models (Zhou et al., 2022; Nie et al., 2022; Wu et al., 2021a; Liu et al., 2023; Chen et al., 2024), with their attention mechanism, can capture long dependencies and extract global information, leading to widespread applications in long-time series prediction. However, this case-by-case paradigm requires meticulous hyperparameter design for different datasets, and its predictive performance can also be affected by data scarcity.

### 2.2. Time Series Forecasting Foundation Model

Pre-training with multiple sources time series has recently received widespread attention (Rasul et al., 2023; Dooley et al., 2024; Garza & Mergenthaler-Canseco, 2023; Kamarthi & Prakash, 2023). MOMENT (Goswami et al., 2024) and MOIRAI (Woo et al., 2024) adopt a BERT-style pre-training approach, while Timer (Liu et al., 2024), Chronos (Ansari et al., 2024) and TimsFM (Das et al.,

2023a) use a GPT-style pre-training approach, giving rise to improved performance in time series prediction. However, the above methods overlook domain-specific information from multi-source data, thus limiting the performance of the models. Different from previous approaches, ROSE pre-trains on large-scale data from various domains and it considers both generalized representations and domain-specific information, which facilitates flexible adaptive transfer in downstream tasks.

## 3. Methodology

### 3.1. Architecture

As illustrated in Figure 2, ROSE adopts an encoder-decoder architecture for time series modeling. Its backbone comprises multiple Transformer layers, which effectively process sequential information and capture temporal dependencies (Vaswani et al., 2017). Both the reconstruction decoder and prediction decoder share the same structure as the Transformer encoder and are designed for reconstruction and prediction tasks, respectively. The reconstruction task enables the model to gain a comprehensive understanding of time series, while the prediction task enhances its few-shot and zero-shot capabilities. ROSE is pre-trained in a channel-independent way, which is widely used in time series forecasting (Nie et al., 2022).

**Input representations.** To enhance the generalization of ROSE for adaptive transferring from multi-domains to different target domains, we model the inputs x with **patch tokens** and **register tokens**. Patch tokens are obtained by partitioning the time series using patching layers (Nie et al., 2022), to preserve local temporal information. Register tokens that capture domain-specific information will be in-

troduced in Section 3.3.

## 3.2. Decomposed Frequency Learning

As shown in Figure 1, time series data are composed of multiple superimposed frequency components, resulting in the overlap of different temporal changes. Furthermore, low-frequency components typically contain information about overall trends and longer-scale variations, and high-frequency components usually contain information about short-term fluctuations and shorter-scale variations, therefore, understanding time series from low and high frequencies separately benefits general time series representation learning. Based on the observations above, we propose a novel frequency-based masked modeling that randomly mask either high-frequency or low-frequency components of a time series multiple times as the key to enable learning of common time series patterns, such as trends and various long and short term fluctuations. Finally, reconstruction task assists the model in comprehending the data from various frequency perspectives, enabling it to learn generalized representations. In contrast, existing frequency masking methods (Zhang et al., 2022), which randomly mask frequencies of a single time series once, show limited forecasting effectiveness due to the lack of common pattern learning from heterogeneous time series that come from various domains.

**Multi-frequency masking.** As shown in the green part of Figure 3, given a time series $\mathbf{x} \in \mathbb{R}^L$, we utilize the Real Fast Fourier Transform (rFFT) (Brigham & Morrow, 1967) to transform it into the frequency domain, giving rise to $\mathbf{x}_{\text{freq}} \in \mathbb{C}^{L/2+1}$.

$$\mathbf{x}_{\text{freq}} = \text{rFFT}(\mathbf{x}). \tag{1}$$

To separately model high-frequency and low-frequency information in time series, we sample $K_{\text{f}}$ thresholds $\tau_1, \tau_2, \tau_3, ..., \tau_{K_{\text{f}}}$ and $K_{\text{f}}$ random numbers $\mu_1, \mu_2, \mu_3, ..., \mu_{K_{\text{f}}}$ for multi-frequency masks, where $\tau \in \text{Uniform}(0, a)$, $a < L/2 + 1$, and $\mu \in \text{Bernoulli}(p)$. Each pair of $\tau_i$ and $\mu_i$ corresponds to the $i_{th}$ frequency mask. This generates a mask matrix $\mathbf{M} \in \{0, 1\}^{K_{\text{f}} \times (L/2+1)}$, where each row corresponds to the $i_{th}$ frequency mask, each column corresponds to the $j_{th}$ frequency, and each element $m_{ij}$ is 0 or 1, meaning that the $j_{th}$ frequency is masked with the $i_{th}$ frequency mask or not.

$$m_{ij} = \begin{cases} \mu_i & , if\ j < \tau_i, \\ (1 - \mu_i) & , if\ j > \tau_i \end{cases}, \tag{2}$$

where $\tau_i$ and $\mu_i$ denote the threshold and random number for the $i_{th}$ frequency domain mask. If $\mu_i = 1$, it means that frequency components above $\tau_i$ will be masked, indicating to mask high frequency , as shown by the threshold $\tau_1$ in Figure 3. Conversely, if $\mu_i = 0$, it signifies that frequency

components below $\tau_i$ will be masked, indicating to mask low frequency, exemplified by threshold $\tau_2$ in Figure 3.

After obtaining the mask matrix $\mathbf{M}$, we replicate $\mathbf{x}_{\text{freq}}$ $K_{\text{f}}$ times to get the $\mathbf{X}_{\text{freq}} \in \mathbb{C}^{K_{\text{f}} \times L/2+1}$ and perform element-wise Hadamard product with the mask matrix $\mathbf{M}$ to get masked frequency of time series. Then, we use the inverse Real Fast Fourier Transform (irFFT) to convert the results from the frequency domain back to the time domain and get $K_{\text{f}}$ masked sequences $\mathbf{X}_{\text{mask}} = \{\mathbf{x}^i_{\text{mask}}\}_{i=1}^{K_{\text{f}}}$, where each $\mathbf{x}^i_{\text{mask}} \in \mathbb{R}^L$ corresponding to masking with a different threshold $\tau_i$.

$$\mathbf{X}_{\text{mask}} = \text{irFFT}(\mathbf{X}_{\text{freq}} \odot \mathbf{M}). \tag{3}$$

**Representation learning.** As shown in the yellow part of Figure 3, after obtaining the $K_{\text{f}}$ masked sequences $\mathbf{X}_{\text{mask}}$, we divide each sequence $\mathbf{x}^i_{\text{mask}}$ into $P$ non-overlapping patches, and use a linear layer to transforming them into $P$ patch tokens, and thus we get $\mathcal{X}_{\text{mp}} = \{\mathbf{X}^i_{\text{mp}}\}_{i=1}^{K_{\text{f}}}$ to capture general information, where each $\mathbf{X}^i_{\text{mp}} \in \mathbb{R}^{P \times D}$, and $D$ is the dimension for each patch token. We replicate the register tokens $\mathbf{X}_{\text{u}}$ $K_{\text{f}}$ times to get $\mathcal{X}_{\text{u}} \in \mathbb{R}^{K_{\text{f}} \times N_{\text{r}} \times D}$, where $\mathbf{X}_{\text{u}}$ is obtained by inputting the original sequence into the TS-Register, as detailed in Section 3.3. Then, we concatenate the patch tokens $\mathcal{X}_{\text{mp}}$ with the register tokens $\mathcal{X}_{\text{u}}$, and feed them into the Transformer encoder to obtain the representation of each masked series. These representations are then averaging aggregated to yield a unified representation $\mathbf{S}_{\text{m}} \in \mathbb{R}^{(N_{\text{r}}+P) \times D}$.

$$\mathbf{S}_{\text{m}} = \text{Aggregator}(\text{Encoder}(\text{Concat}(\mathcal{X}_{\text{mp}}, \mathcal{X}_{\text{u}}))). \tag{4}$$

**Reconstruction task.** After obtaining the representation $\mathbf{S}_{\text{m}}$, we feed it into the reconstruction decoder, which shares same stucture as the Tranformer encoder, and ultimately reconstruct the original sequence $\hat{\mathbf{x}} \in \mathbb{R}^L$ through the reconstruction head, which is a linear layer. As frequency domain masking affects the overall time series, we compute the Mean Squared Error (MSE) reconstruction loss for the entire time series.

$$\mathcal{L}_{\text{reconstruction}} = ||\mathbf{x} - \hat{\mathbf{x}}||_2^2. \tag{5}$$

## 3.3. Time Series Register

By decomposed frequency learning, we can obtain the general representations. Additionally, we propose the TS-Register that learns domain-specific information from the multi-domain datasets for adaptive transfer. It clusters domain-specific information from the multi-domain datasets into register tokens and stores such domain-specific information in the register during pre-training. Then, it adaptively selects domain-specific information from the register via a Top-K selection strategy to enhance the performance in the

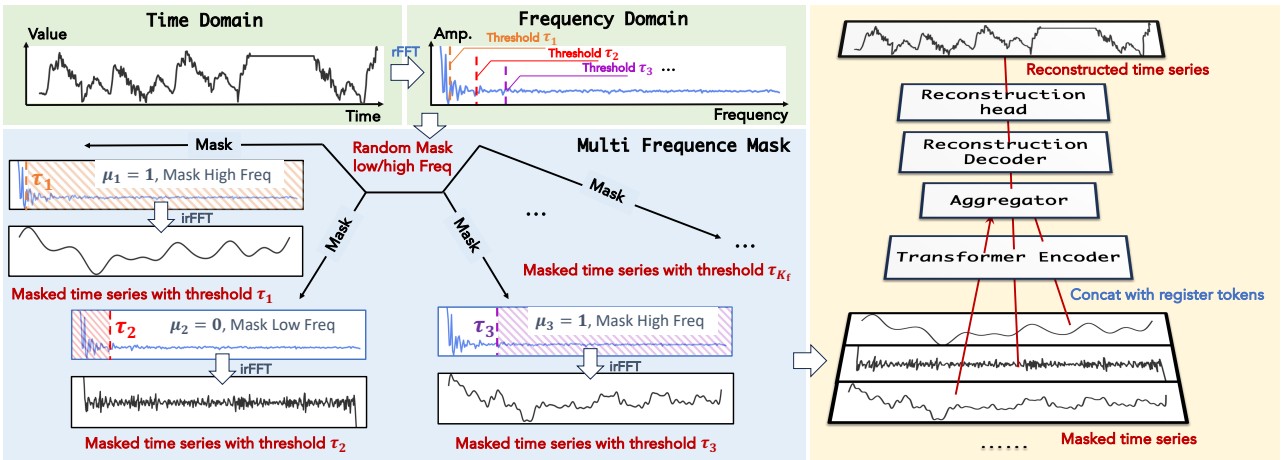

*Figure 3.* An illustration of decomposed frequency learning. Based on the sampled thresholds, we randomly apply low/high-frequency masking to the time series in the frequency domain and then transform it back to the time domain for reconstruction.

target domain. A novel learnable low-rank matrix is proposed to set to complement the downstream dataset-specific information through fine-tuning.

We set up a randomly initialized register $\mathbf{E} \in \mathbb{R}^{H \times D_r}$ with $H$ cluster center vectors $\mathbf{e}_i \in \mathbb{R}^{D_r}, i \in \{1, 2, \ldots, H\}$. Each of input time series $\mathbf{x} \in \mathbb{R}^L$ is projected into a data-dependent embedding $\mathbf{x}_e \in \mathbb{R}^{D_r}$ through a linear layer.

**Pre-training stage.** As shown in Figure 2(b), we use the register to cluster these data-dependent embeddings, which generate domain-specific information, and store them in pre-training. Specifically, We find a cluster center vector $\mathbf{e}_\delta$ from the register $\mathbf{E}$ where we use $\delta$ to denote the cluster that the data-dependent embedding $\mathbf{x}_e$ belongs to.

$$\mathcal{L}_{\text{register}} = \|\mathbf{x}_e - \mathbf{e}_\delta\|_2^2, \quad \delta = \arg\min_{j=1:H} \|\mathbf{x}_e - \mathbf{e}_j\|_2. \quad (6)$$

To update the cluster center vectors in the register $\mathbf{E}$ that represents the domain information of the pre-trained datasets, we set the loss function shown in Equation 6 that minimizes the distance between the embedding $\mathbf{x}_e$ and the cluster center $\mathbf{e}_\delta$. To solve the problem that the gradient of the $\arg\min$ function cannot be backpropagated, we use the stop gradient operation to pass the gradient of $\mathbf{e}_\delta$ directly to $\mathbf{x}_e$.

In this way, the vectors in the register $\mathbf{E}$ cluster the embeddings of different data and learn the domain-specific centers for pre-trained datasets, which can represent domain-specific information. As a vector in the register $\mathbf{E}$, $\mathbf{e}_\delta$ represents the domain-specific information for input $\mathbf{x}$. $\mathbf{e}_\delta$ is invariant under small perturbations in $\mathbf{x}_e$ that represents $\mathbf{x}$, which promotes better representation of domain-specific information and robustness of the vectors in the register. This also avoids their over-reliance on detailed information about specific datasets.

The cluster center vector $\mathbf{e}_\delta$ is then patched into $\mathbf{X}_u \in \mathbb{R}^{N_r \times D}$, where $N_r$ is the number of the register tokens and $D$ is the dimensionality of Transformer latent space. $\mathbf{X}_u$ is called register tokens, which are used as the prefix of the patch tokens $\mathbf{X}_p \in \mathbb{R}^{P \times D}$ and input for the Transformer encoder to provide domain-specific information.

**Fine-tuning stage.** As shown in Figure 2(c), after obtaining a register $\mathbf{E}$ that contains domain-specific information through pre-training, we freeze the register parameters to adaptively use this domain-specific information in the downstream tasks.

Since the target domain may not strictly fit one of the upstream domains, we propose a novel embedding learning of the downstream data by employing a Top-K strategy that selects $k$ similar vectors in the register. As shown in Equation 7, the embedding of input time series $\mathbf{x}_e$ picks the $k$ nearest vectors in the register $\mathbf{E}$, and uses their average as $\bar{\mathbf{e}}_k$ to represent the domain-specific information from the pre-train stage. $\bar{\mathbf{e}}_k$ is also patched into $\mathbf{X}_d \in \mathbb{R}^{N_r \times D}$ and is used as the **domain specific register tokens**.

$$\bar{\mathbf{e}}_k = \frac{1}{k} \sum_{i=1}^{k} \mathbf{e}_{\delta_i}, \quad \{\delta_1, \cdots, \delta_k\} = \arg\text{Topk}_{j=1:H}\left(\frac{1}{\|\mathbf{x}_e - \mathbf{e}_j\|_2}\right). \quad (7)$$

Since the downstream data has its own specific information at the dataset level in addition to the domain level, this may not be fully represented by the domain information obtained from the pre-trained dataset alone. Therefore, we innovatively set a learnable matrix $\mathbf{A} \in \mathbb{R}^{N_r \times D}$ to adjust $\mathbf{X}_d$ to complement the **specific information of downstream data**. Since the pre-trained model has a very low intrinsic dimension (Aghajanyan et al., 2020), in order to get better fine-tuning results, $\mathbf{A}$ is set as a low-rank matrix:

$$\mathbf{A} = \mathbf{u} \times \mathbf{v}^{\text{T}}, \quad (8)$$

where $\mathbf{u} \in \mathbb{R}^{N_r}$ and $\mathbf{v} \in \mathbb{R}^D$, and only the vectors $\mathbf{u}$ and $\mathbf{v}$ need to be retrained in the fine-tuning step. As illustrated

in Equation 9, the register token $\mathbf{X}_r$ of the downstream scenario is obtained by doing the Hadamard product of $\mathbf{X}_d$, which represents the domain-specific information obtained at the pre-train stage, and $\mathbf{A}$, which represents the downstream dataset-specific information.

$$\mathbf{X}_r = \mathbf{X}_d \odot \mathbf{A}. \tag{9}$$

### 3.4. Training

To improve the prediction performance in zero-shot and few-shot settings, we co-train supervised prediction with self-supervised reconstruction that uses multi-frequency masking to learn unified features that are more applicable to the downstream prediction task.

**Prediction task.** The input time series $\mathbf{x} \in \mathbb{R}^L$ is sliced into $P$ non-overlapping patches and then mapped to $\mathbf{X}_p \in \mathbb{R}^{P \times D}$. Based on common forecasting needs (Qiu et al., 2024), we set up four prediction heads mapping to prediction lengths of $\{96, 192, 336, 720\}$ to accomplish the prediction task. Patch tokens $\mathbf{X}_p$ are concatenated with the register tokens $\mathbf{X}_u$ and then successively fed into the Transformer encoder to yield the representation $\mathbf{S} \in \mathbb{R}^{(N_r+P) \times D}$:

$$\mathbf{S} = \text{Encoder}(\text{Concatenate}(\mathcal{X}_p, \mathcal{X}_u)). \tag{10}$$

We feed the representation $\mathbf{S}$ into the prediction decoder and prediction heads to obtain four prediction results $\hat{\mathbf{Y}}_F$, where $F \in \{96, 192, 336, 720\}$. With the ground truth $\mathbf{Y}_F$, the prediction loss $\mathcal{L}_{\text{prediction}}$ is shown in Equation 11.

$$\mathcal{L}_{\text{prediction}} = \sum_{F \in \{96,192,336,720\}} ||\mathbf{Y}_F - \hat{\mathbf{Y}}_F||_2^2. \tag{11}$$

**Pre-training.** The reconstruction task learns generalized features through the Transformer encoder and reconstruction decoder. To utilize these features for the prediction task, the parameters of the reconstruction decoder are copied to the prediction decoder during forward propagation. To avoid prediction training affecting the generalization performance of the model, the gradients of the prediction heads are skipped at back-propagation. The overall loss of ROSE in pre-training stage is shown in Equation 12.

$$\mathcal{L}_{\text{pre-train}} = \mathcal{L}_{\text{reconstruction}} + \mathcal{L}_{\text{prediction}} + \mathcal{L}_{\text{register}}. \tag{12}$$

**Fine-tuning.** We only perform a prediction task in fine-tuning. Patch tokens $\mathbf{X}_p$ are concatenated with the adjusted register tokens $\mathbf{X}_r$. For a downstream task with a fixed prediction length, we use the corresponding pre-trained prediction head to fine-tune the model.

## 4. Experiments

**Pre-training datasets.** The datasets are crucial for pre-training a general time series forecasting model. In light of this, we gather many publicly available datasets from various domains, including energy, nature, health, transport, web, economics, etc. The details of these datasets are shown in the Appendix A.1.1. To enhance data utilization, we downsample fine-grained datasets to coarser granularity, resulting in approximately 887 million time points.

**Evaluation datasets.** To conduct comprehensive and fair comparisons for different models, we conduct experiments on seven well-known forecasting benchmarks as the target datasets, including Weather, Traffic, Electricity, and ETT (4 subsets), which cover multiple domains.

**Baselines.** We select the state-of-the-art models as baselines in full-shot and few-shot setting, including four specific models: iTransformer (Liu et al., 2023), PatchTST (Nie et al., 2022), TimesNet (Wu et al., 2022), and DLinear (Zeng et al., 2023), and two LLM-based models: GPT4TS (Zhou et al., 2024) and $S^2$IP-LLM (Pan et al., 2024). In addition, we select five foundation models for comparison in zero-shot setting, including Timer (Liu et al., 2024), MOIRAI (Woo et al., 2024), Chronos (Ansari et al., 2024), TimesFM (Das et al., 2023b), and Moment (Goswami et al., 2024).

**Setup.** Consistent with previous works, we adopted Mean Squared Error (MSE) and Mean Absolute Error (MAE) as evaluation metrics. Due to ROSE mostly aims at long-term predictions, for fair comparison, all methods fix the look-back window $L = 512$ and predict the future values with lengths $F = \{96, 192, 336, 720\}$. More implementation details are presented in the Appendix A.1.3.

### 4.1. In Distribution Forecasting

**Setting.** In full-shot setting, we utilize full downstream data to fine-tune pre-trained ROSE and baselines. In few-shot setting, we fine-tune all models with only 10% train data. The *"Drop Last"* issue is reported by several researchers (Qiu et al., 2024; 2025a; Li et al., 2025). That is, in some previous works evaluating the model on test set with drop-last=True setting may cause additional errors related to test batch size. In our experiment, to ensure fair comparison in the future, we set the drop last to False for all baselines to avoid this issue.

**Full-shot results.** As shown in Table 1, we also present the results of the ROSE in 10% few-shot setting. Key observations are summarized as follows. First, as a general forecasting model, ROSE *achieves superior performance compared to the six state-of-the-art baselines with full-data training, achieving an average MSE reduction of 15%,* which shows that our decomposed frequency learning and register help to learn generalized representations from large-scale datasets and adaptively transfer the multi-domain information to specific downstream scenarios. Second, we observe that

ROSE in 10% few-shot setting shockingly *improves a large margin as MSE reduction in average exceeding 12% over the baselines trained with full data.* This observation validates the transferability of ROSE pre-trained with large multi-source data.

**Few-shot results.** The results under the 10% few-shot setting are presented in Table 13 in Appendix A.10.2. ROSE outperforms advanced models when training data is scarce in the target domain. Figure 4 shows the performance of pre-trained ROSE and ROSE trained from scratch on ETTh1 and ETTm2 with different fine-tuning data percentages, noting the best baselines in full-shot setting. The pre-trained ROSE shows stable, superior performance even with limited fine-tuning samples. Specifically, the pre-trained ROSE *exceeds SOTA performance with only 1% train data for ETTh1 and 2% for ETTm2.* Moreover, compared to the ROSE trained from scratch, the pre-trained ROSE exhibits a slower decline in prediction performance with the reduction of fine-tuning data, demonstrating the impressive generalization ability of ROSE through pre-training.

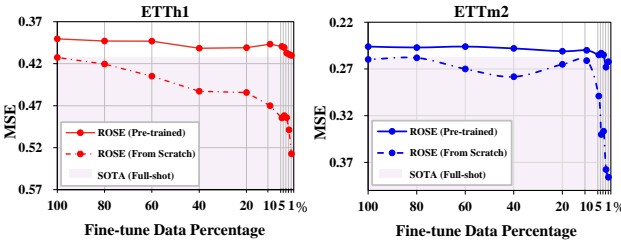

*Figure 4.* The forecasting results of ROSE obtained by training from scratch and fine-tuning from the pre-trained model. The right, upper corner is the best case.

### 4.2. Zero-shot Forecasting

**Setting.** In this section, to ensure a fair comparison, we conduct zero-shot predictions for each foundational model on downstream datasets not included in their pre-training data. It is worth noting that, unlike a few foundation models (Woo et al., 2024) that require much longer inputs to achieve better predictive performance, we fix the input length of all baselines to 512 without considering longer input lengths, as many real-world scenarios could offer very limited samples.

**Results.** As shown in Table 2, ROSE significantly outperforms across the majority of datasets, *achieving an average reduction of 15% in MSE.* In comparison to Timer and Moirai, ROSE achieves average MSE reductions of 9% and 6%, respectively, and demonstrates a remarkable 43% relative improvement over Moment. Notably, ROSE stands out not only for its superior performance but also for its exceptionally lightweight and efficient design, which sets it apart from other foundational models. Detailed analysis of these aspects will be presented in Section 4.3.

### 4.3. Model Analysis

**Efficiency analysis.** To exhibit the performance and efficiency advantages of ROSE, we compare its parameter count to other foundation models and evaluate their performance and testing time averaged on ETTh1 and ETTh2 datasets in zero-setting. Similarly, for each specific model, we evaluate its parameter count as well as its performance in full-shot setting and training-to-testing time averaged on the same datasets. Specific implementation details and results can be found in Appendix A.2.

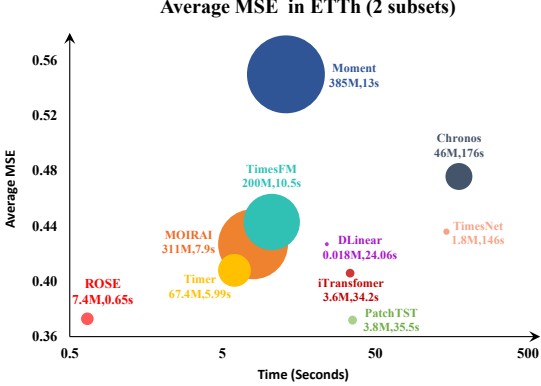

*Figure 5.* Model performance, number of parameter and efficiency comparison.

As shown in Figure 5, ROSE is a lightweight general model with 7.4M parameters and short inference time, which are only about one-tenth of the second fastest/smallest foundation model (Timer). Importantly, ROSE uses the least number of parameters among foundation models, with its parameter count approaching that of specific models, while exhibiting superior zero-shot performance. This is attributed to our proposed decomposed frequency learning that enhances the comprehension of time series. Concurrently, the TS-Register achieves the adaptive transfer thus efficiently adapting to downstream tasks without the need of scaling up to achieve strong generalizability. Compared to foundation models with large scale, ROSE may better meet the need for general models in real scenarios that require high computational and parameter efficiency as well as high prediction accuracy with scarce downstream data.

**Visualization of TS-Register.** To validate the TS-Register's capability to transfer domain-specific information adaptively from pre-training datasets to target datasets, we visualize the cosine similarity of register vector selections from datasets across different domains. As shown in Figure 6(a), the cosine similarity is higher for datasets within the same domain and lower between different domains. We also visualize the register vector selections from different datasets in Figures 6(b) and (c), where datasets from the same domain show similar visualizations. This confirms the TS-Register's capability of adaptive transfer from multi-source to target datasets across various domains.

*Table 1.* The results for ROSE in full-shot setting and 10% few-shot setting, compared with other methods in full-shot setting. The average results of all predicted lengths are listed here.

| Models | ROSE | | ROSE (10%) | | ITransformer | | PatchTST | | Timesnet | | Dlinear | | GPT4TS | | S$^2$IP-LLM | |
|---|---|---|---|---|---|---|---|---|---|---|---|---|---|---|---|---|
| Metric | MSE | MAE | MSE | MAE | MSE | MAE | MSE | MAE | MSE | MAE | MSE | MAE | MSE | MAE | MSE | MAE |
| ETTh1 | **0.391** | **0.414** | 0.397 | 0.419 | 0.439 | 0.448 | 0.413 | 0.434 | 0.582 | 0.533 | 0.416 | 0.436 | 0.427 | 0.426 | 0.406 | 0.427 |
| ETTh2 | **0.331** | **0.374** | 0.335 | 0.380 | 0.374 | 0.406 | **0.331** | 0.381 | 0.409 | 0.438 | 0.508 | 0.485 | 0.354 | 0.394 | 0.347 | 0.391 |
| ETTm1 | **0.341** | **0.367** | 0.349 | 0.372 | 0.362 | 0.391 | 0.353 | 0.382 | 0.490 | 0.464 | 0.356 | 0.378 | 0.352 | 0.383 | 0.343 | 0.379 |
| ETTm2 | **0.246** | **0.305** | 0.250 | 0.308 | 0.269 | 0.329 | 0.256 | 0.317 | 0.317 | 0.358 | 0.259 | 0.325 | 0.266 | 0.326 | 0.257 | 0.319 |
| Weather | **0.217** | **0.251** | 0.224 | 0.252 | 0.233 | 0.271 | 0.226 | 0.264 | 0.329 | 0.336 | 0.239 | 0.289 | 0.237 | 0.270 | 0.222 | 0.259 |
| Electricity | **0.155** | **0.248** | 0.164 | 0.253 | 0.164 | 0.261 | 0.159 | 0.253 | 0.195 | 0.296 | 0.166 | 0.267 | 0.167 | 0.263 | 0.161 | 0.257 |
| Traffic | **0.390** | **0.264** | 0.418 | 0.278 | 0.397 | 0.282 | 0.391 | **0.264** | 0.623 | 0.333 | 0.433 | 0.305 | 0.414 | 0.294 | 0.405 | 0.286 |

*Table 2.* The results for ROSE and other foundation models in the zero-shot setting. The average results of all predicted lengths are listed here. We use '-' to indicate that the dataset has been involved in the model's pre-training, and thus not used for testing.

| Models | ROSE | Timer | MOIRAI | Chronos | TimesFM | Moment |
|---|---|---|---|---|---|---|
| Metric | MSE | MSE | MSE | MSE | MSE | MSE |
| ETTh1 | **0.401** | 0.451 | 0.475 | 0.560 | 0.489 | 0.708 |
| ETTh2 | **0.346** | 0.366 | 0.379 | 0.392 | 0.396 | 0.392 |
| ETTm1 | 0.525 | 0.544 | 0.714 | 0.636 | **0.434** | 0.697 |
| ETTm2 | **0.299** | 0.360 | 0.343 | 0.313 | 0.320 | 0.319 |
| Weather | **0.265** | 0.292 | 0.267 | 0.288 | - | 0.291 |
| Electricity | **0.234** | 0.297 | 0.241 | 0.245 | - | 0.861 |
| Traffic | **0.588** | 0.613 | - | 0.615 | - | 1.411 |

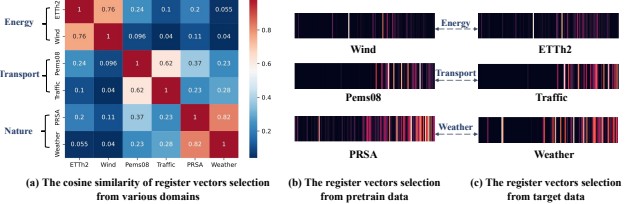

(a) The cosine similarity of register vectors selection from various domains   (b) The register vectors selection from pretrain data   (c) The register vectors selection from target data

*Figure 6.* Visualization of TS-Register. The calculation of cosine similarity is in the Appendix A.3.

**Scalability and sensitivity.** The scalability analysis of ROSE's model size and pre-training data size are presented in Appendix A.4. The sensitivity analyses for the upper bound $a$ of the thresholds, the number of masked series $K_f$, the number of register tokens $N_r$, the size of register $H$ and number of selections $k$ in Top-K strategy are presented in Appendix A.5.

## 4.4. Ablation Studies

**Model architecture.** To validate effectiveness of our model design, we perform ablation studies on TS-Register, prediction tasks, and reconstruction task in 10% few-shot setting. Table 3 shows the impact of each module. The TS-Register leverages multi-domain information during pre-training, aiding adaptive transfer to downstream datasets, as further discussed in Section 4.3. The prediction tasks enhance performance in data-scarce situations. Without it, performance

significantly drops on ETTh1 and ETTh2 with limited samples. Without the reconstruction task, our model shows negative transfer effects on ETTm1 and ETTm2, likely due to the prediction task making the model more susceptible to pre-training data biases.

**Masking method.** To further validate the effectiveness of decomposed frequency learning, we replace the multi-frequency masking with different masking methods, including two mainstream time-domain methods: patch masking (Nie et al., 2022) and multi-patch masking (Dong et al., 2024), as well as random frequency masking (Chen et al., 2023b). The results in Table 4 show that random frequency masking and patch masking led to negative transfer on ETTm1 and ETTm2, likely due to significant disruption of the original time series, causing overfitting. In contrast, multi-patch masking and multi-frequency masking resulted in positive transfer across all datasets by preventing excessive disruption. Multi-frequency masking achieved better results, demonstrating its ability to help the model understand temporal patterns from a multi-frequency perspective. We also compare with some other pre-training tasks in Table 14 in Appendix A.10.3.

*Table 3.* Ablations on key components of model architecture, including TS-register, prediction task and reconstruction task. The average results of all predicted lengths are listed here.

| Design | | ETTm1 | | ETTm2 | | ETTh1 | | ETTh2 | |
|---|---|---|---|---|---|---|---|---|---|
| | | MSE | MAE | MSE | MAE | MSE | MAE | MSE | MAE |
| ROSE | | **0.349** | **0.372** | **0.250** | **0.308** | **0.397** | **0.419** | **0.335** | **0.380** |
| w/o | TS-Register | 0.354 | 0.378 | 0.256 | 0.312 | 0.418 | 0.427 | 0.355 | 0.390 |
| | Prediction Task | 0.360 | 0.384 | 0.257 | 0.314 | 0.422 | 0.438 | 0.372 | 0.410 |
| | Reconstruction Task | 0.387 | 0.403 | 0.269 | 0.327 | 0.412 | 0.428 | 0.361 | 0.399 |
| From scratch | | 0.371 | 0.391 | 0.261 | 0.318 | 0.470 | 0.480 | 0.400 | 0.425 |

*Table 4.* Ablations on decomposed frequency learning, where we replace Multi-freq masking with other masking methods. The average results of all predicted lengths are listed here.

| Design | ETTm1 | | ETTm2 | | ETTh1 | | ETTh2 | |
|---|---|---|---|---|---|---|---|---|
| | MSE | MAE | MSE | MAE | MSE | MAE | MSE | MAE |
| ROSE | **0.349** | **0.372** | **0.250** | **0.308** | **0.397** | **0.419** | **0.335** | **0.380** |
| Random Freq Masking | 0.381 | 0.397 | 0.261 | 0.324 | 0.410 | 0.427 | 0.374 | 0.405 |
| Multi-Patch Masking | 0.356 | 0.379 | 0.259 | 0.316 | 0.404 | 0.426 | 0.349 | 0.389 |
| Patch Masking | 0.378 | 0.400 | 0.261 | 0.319 | 0.408 | 0.432 | 0.375 | 0.407 |
| From scratch | 0.371 | 0.391 | 0.261 | 0.318 | 0.470 | 0.480 | 0.400 | 0.425 |

# 5. Conclusion and Future Work

In this work, we propose ROSE, a novel general model, addressing the challenges of leveraging multi-domain datasets for enhancing downstream prediction task performance. ROSE utilizes decomposed frequency learning and TS-Register to capture generalized and domain-specific representations, enabling improved fine-tuning results, especially in data-scarce scenarios. Our experiments demonstrate ROSE's superior performance over baselines with both full-data and few-data fine-tuning, as well as its impressive zero-shot capabilities. Future efforts will concentrate on expanding pre-training datasets and extending ROSE's applicability across diverse time series analysis tasks, e.g., classification. We provide our code at https://github.com/decisionintelligence/ROSE.

# Impact Statement

This paper presents work whose goal is to advance the field of Machine Learning. There are many potential societal consequences of our work, none which we feel must be specifically highlighted here.

# Acknowledgements

This work was partially supported by National Natural Science Foundation of China (62372179). Chenjuan Guo is the corresponding author of the work.

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

# A. Appendix

## A.1. Implementation Details

### A.1.1. PRE-TRAINING DATASETS

We use multi-source datasets in pre-training which contain subsets of Monash (Godahewa et al., 2021), UEA (Bagnall et al., 2018) and UCR (Dau et al., 2019) time series datasets, as well as some other time series classical datasets (Zhang et al., 2017; Wang et al., 2024; Liu et al., 2022a; McCracken & Ng, 2016; Taieb et al., 2012). The final list of all pre-training datasets is shown in Table 5. There is no overlap between the pre-training datasets and the target datasets. It is worth noting that the dataset weather in the pre-training dataset is a univariate dataset, which is different to the multivariate dataset weather in the target task. The pre-trained datasets can be categorized into 6 different domains according to their sources: Energy, Nature, Health, Transport, and Web. The sampling frequencies of the datasets show a remarkable diversity, ranging from millisecond samples to monthly samples, which reflects the diverse application scenarios and complexity of the real world. For all pre-training datasets, we split them into univariate sequences and train them in a channel-independent manner.

*Table 5.* List of pretraining datasets.

| Domain | Dataset | Frequency | Time Pionts | Source |
|---|---|---|---|---|
| Energy | Aus. Electricity Demand | Half Hourly | 1155264 | Monash(Godahewa et al., 2021) |
| | Wind | 4 Seconds | 7397147 | Monash(Godahewa et al., 2021) |
| | Wind Farms | Minutely | 172178060 | Monash(Godahewa et al., 2021) |
| | Solar | 10 Minutes | 7200720 | Monash(Godahewa et al., 2021) |
| | Solar Power | 4 Seconds | 7397222 | Monash(Godahewa et al., 2021) |
| | London Smart Meters | Half Hourly | 166527216 | Monash(Godahewa et al., 2021) |
| Nature | Phoneme | - | 2160640 | UCR(Dau et al., 2019) |
| | EigenWorms | - | 27947136 | UEA(Bagnall et al., 2018) |
| | PRSA | Hourly | 4628448 | (Zhang et al., 2017) |
| | Temperature Rain | Daily | 23252200 | Monash(Godahewa et al., 2021) |
| | StarLightCurves | - | 9457664 | UCR(Dau et al., 2019) |
| | Worms | 0.033 Seconds | 232200 | UCR(Dau et al., 2019) |
| | Saugeen River Flow | Daily | 23741 | Monash(Godahewa et al., 2021) |
| | Sunspot | Daily | 73924 | Monash(Godahewa et al., 2021) |
| | Weather | Daily | 43032000 | Monash(Godahewa et al., 2021) |
| | KDD Cup 2018 | Daily | 2942364 | Monash(Godahewa et al., 2021) |
| | US Births | Daily | 7305 | Monash(Godahewa et al., 2021) |
| Health | MotorImagery | 0.001 Seconds | 72576000 | UEA(Bagnall et al., 2018) |
| | SelfRegulationSCP1 | 0.004 Seconds | 3015936 | UEA(Bagnall et al., 2018) |
| | SelfRegulationSCP2 | 0.004 Seconds | 3064320 | UEA(Bagnall et al., 2018) |
| | AtrialFibrillation | 0.008 Seconds | 38400 | UEA(Bagnall et al., 2018) |
| | PigArtPressure | - | 624000 | UCR(Dau et al., 2019) |
| | PIGCVP | - | 624000 | UCR(Dau et al., 2019) |
| | TDbrain | 0.002 Seconds | 79232703 | (Wang et al., 2024) |
| Transport | Pems03 | 5 Minute | 9382464 | (Liu et al., 2022a) |
| | Pems04 | 5 Minute | 5216544 | (Liu et al., 2022a) |
| | Pems07 | 5 Minute | 24921792 | (Liu et al., 2022a) |
| | Pems08 | 5 Minute | 3035520 | (Liu et al., 2022a) |
| | Pems-bay | 5 Minute | 16937700 | (Liu et al., 2022a) |
| | Pedestrian_Counts | Hourly | 3132346 | Monash(Godahewa et al., 2021) |
| Web | Web Traffic | Daily | 116485589 | Monash(Godahewa et al., 2021) |
| Economic | FRED_MD | Monthly | 77896 | (McCracken & Ng, 2016) |
| | Bitcoin | Daily | 75364 | Monash(Godahewa et al., 2021) |
| | NN5 | Daily | 87801 | (Taieb et al., 2012) |

### A.1.2. EVALUATION DATASETS

We use the following 7 multivariate time-series datasets for downstream fine-tuning and forecasting: ETT datasets[1] contain 7 variates collected from two different electric transformers from July 2016 to July 2018. It consists of four subsets, of which ETTh1/ETTh2 are recorded hourly and ETTm1/ETTm2 are recorded every 15 minutes. Traffic[2] contains road occupancy rates measured by 862 sensors on freeways in the San Francisco Bay Area from 2015 to 2016, recorded hourly. Weather[3] collects 21 meteorological indicators, such as temperature and barometric pressure, for Germany in 2020, recorded every 10 minutes. Electricity[4] contains the electricity consumption of 321 customers from July 2016 to July 2019, recorded hourly. We split each evaluation dataset into train-validation-test sets and detailed statistics of evaluation datasets are shown in Table 6.

*Table 6.* The statistics of evaluation datasets.

| Dataset | ETTm1 | ETTm2 | ETTh1 | ETTh2 | Traffic | Weather | Electricity |
|---|---|---|---|---|---|---|---|
| Variables | 7 | 7 | 7 | 7 | 862 | 21 | 321 |
| Timestamps | 69680 | 69680 | 17420 | 17420 | 17544 | 52696 | 26304 |
| Split Ratio | 6:2:2 | 6:2:2 | 6:2:2 | 6:2:2 | 7:1:2 | 7:1:2 | 7:1:2 |

### A.1.3. SETTING

We implemented ROSE in PyTorch (Paszke et al., 2019) and all the experiments were conducted on 8 NVIDIA A800 80GB GPU. We used ADAM (Kingma & Ba, 2014) with an initial learning rate of $5 \times 10^{-4}$ and implemented learning rate decay using the StepLR method to implement learning rate decaying pre-training. By default, ROSE contains 3 encoder layers and 3 decoder layers with head number of 16 and the dimension of latent space $D = 256$. The patch size for patching is set to 64.

**Pre-training.** We use $N_r = 3$ as the number of register tokens and $P = 8$ as the path tokens. We set the input length to 512 for the supervised prediction task with target lengths of 96, 192, 336, and 720. We also set the input length to 512 and mask number $K_f = 4$. The batch size is set to 8192 in pre-training.

**Fine-tuning.** We fix the lookback window to 512, and perform predictions with target lengths of 96, 192, 336, and 720, respectively. The number of register tokens $N_r$ and patch tokens $P$ is the same as in pre-training, and the parameter $k = 3$ in TopK is set when selection vectors are performed in the register.

**The t-SNE visualization.** We select three datasets (Pems08, PSRA, Electricity) from transport, nature and energy domains respectively and compare the differences in hidden representations between direct transfer and adaptive transfer. Specifically, direct transfer refers to the case where domain specific information is not considered, while adaptive transfer considers domain specific information that is learned by register tokens. We visualized the output of the encoder's hidden representations using t-SNE.

### A.1.4. BASELINES

We select the state-of-the-art models as our baselines in full-shot and few-shot setting, including four specific models: iTransformer (Liu et al., 2023), PatchTST (Nie et al., 2022), TimesNet (Wu et al., 2022), and DLinear (Zeng et al., 2023), and two LLM-based models: GPT4TS (Zhou et al., 2024) and S$^2$IP-LLM (Pan et al., 2024). In addition, we selected five foundation models for comparison in zero-shot setting, including Timer (Liu et al., 2024), MOIRAI (Woo et al., 2024), Chronos (Ansari et al., 2024), TimesFM (Das et al., 2023b) and Moment (Goswami et al., 2024). The zero-shot experiment of Moment is designed based on the reconstruction task in the pre-training phase. Specifically, to ensure consistency between pre-training (reconstruction) and downstream prediction tasks, we actively mask the time periods to be predicted in the input sequence, and directly use the model's reconstruction output for this part as the prediction value. Moment itself is not designed for zero-shot prediction and does not officially support zero-shot forecasting in this manner. The specific code base for these models is listed in Table 7:

---

[1] https://github.com/zhouhaoyi/ETDataset

[2] https://pems.dot.ca.gov/

[3] https://www.bgc-jena.mpg.de/wetter/

[4] https://archive.ics.uci.edu/ml/datasets/ElectricityLoadDiagrams20112014

*Table 7.* Code repositories for baselines.

| Model Types | Models | Code Repositories |
|---|---|---|
| Small Model | iTransformer | https://github.com/thuml/iTransformer |
| | PatchTST | https://github.com/yuqinie98/PatchTST |
| | TimesNet | https://github.com/thuml/TimesNet |
| | Dlinear | https://github.com/cure-lab/LTSF-Linear |
| Foundation Model | Timer | https://github.com/thuml/Large-Time-Series-Model |
| | MOIRAI | https://github.com/redoules/moirai |
| | Chronos | https://github.com/amazon-science/chronos-forecasting |
| | TimesFM | https://github.com/google-research/timesfm/ |
| | Moment | https://anonymous.4open.science/r/BETT-773F/README.md |
| LLM-based Model | GPT4TS | https://github.com/DAMO-DI-ML/NeurIPS2023-One-Fits-All |
| | S2IP-LLM | https://github.com/panzijie825/S2IP-LLM |

## A.2. Efficiency Analysis

As an important aspect of foundation models, inference efficiency is crucial. Therefore, we evaluate the testing time of ROSE and five foundation models in the ETTh1 and ETTh2 dataset in zero-shot setting. Similarly, we evaluate the time of the entire process of training, validation, and testing for four specific models in the same datasets in full-shot setting. The above experiments all set the batch size to 32. The specific results are shown in Table 8 and Figure 5. We observe that ROSE maintains its advantage in zero-shot performance while also being significantly faster compared to the baselines, even being approximately ten times faster than the second-fastest foundation model, Timer. This raises our reflection on whether time-series foundation models require extremely large parameter sizes and whether existing time-series foundation models have validated their architectures' scaling laws on time-series data.

*Table 8.* Efficiency analysis.

| Model | Parameters | Pre-train datasize | Averaged time |
|---|---|---|---|
| ROSE | 7.4M | 0.89B | 0.652s |
| MOIRAI | 311M | 27B | 7.920s |
| Timer | 67.4M | 1B | 5.989s |
| Chronos | 46M | 84B | 176s |
| TimesFM | 200M | 100B | 10.5s |
| Moment | 385M | 1.13B | 13s |
| Itransformer | 3.8M | - | 34.18s |
| PatchTST | 3.2M | - | 35.47s |
| TimesNet | 1.8M | - | 146s |
| Dlinear | 0.018M | - | 24.06s |

## A.3. Calculation of Cosine Similarity

In Figure 6, we visualize the cosine similarity of register vector selections. For each sample in a dataset, during the inference process, $k$ vectors are selected from the register based on the Top-K strategy. We iterate through all samples in the dataset and count the number of times each vector is selected, which allows us to obtain a record vector of length equivalent to the size of the register for each dataset. The $i_{th}$ position in the record vector represents the number of times the $i_{th}$ vector in the register has been selected by samples in the dataset. For a pair of datasets, we can obtain a unique record vector for each dataset, and then we are able to calculate the cosine similarity of two vectors.

## A.4. Scalability

Scalability is crucial for a general model, enabling significant performance improvements by expanding pre-training data and model sizes. To investigate the scalability of ROSE, we increased both the model size and dataset size and evaluated its

predictive performance on four ETT datasets.

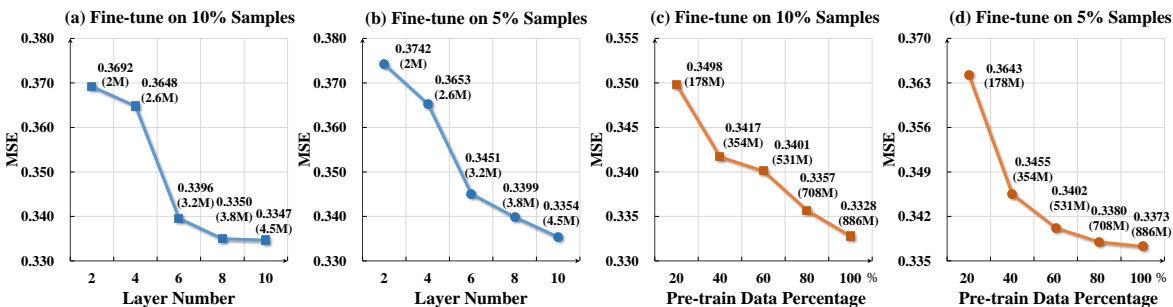

*Figure 7.* (a)/(b): Larger ROSE demonstrates better performance on downstream forecasting. (c)/(d): ROSE pre-trained on larger datasets demonstrates better performance on downstream forecasting.

**Model size.** Constrained by computational resources, we use 40% pre-training datasets. The results are shown in Figure 7(a) and (b). When maintaining the model dimension, we increased the model layers, increasing model parameters from 2M to 4.5M. This led to 10.37% and 9.34% improvements in the few-shot scenario with 5% and 10% downstream data, respectively.

**Data size.** When keeping the model size, we increase the size of the pre-training datasets from 178M to 887M. The results are shown in Figure 7(c) and (d). The performance of our model steadily improves with the increase in dataset size and achieves improvements of 7.4% and 4.8% respectively.

## A.5. Sensitivity

We perform the sensitivity analyses for the upper bound $a$ of the thresholds, the number of masked series $K_f$, the number of register tokens $N_r$, the size of register $H$ and the number of selections $k$ in Top-K strategy. All the sensitivity experiments present the average results on the four ETT datasets: ETTh1, ETTh2, ETTm1 and ETTm2 under 10% few-shot setting.

**Number of masked series.** As described in Section 3.2, we propose decomposed frequency learning, which employs multiple thresholds to randomly mask high and low frequencies in the frequency domain, thereby decomposing the original time series into multiple frequency components. This allows the model to understand the time series from multiple frequency perspectives. In this experiment, we study the influence of the number of masked series $K_f$ on downstream performance. We train ROSE with 1, 2, 3, 4, 5, or 6 mask series. We report the results of this analysis in Figure 8(a). We find that as the number of masked sequences increases, the downstream performance gradually improves. This is because the model can better understand the time series from the decomposed frequency components, which enhances the model's generalization ability. However, more masked series do not bring better downstream performance. This could be due to an excessive number of masked sequences leading to information redundancy. In all our experiments, we keep 4 mask series.

**Number of register tokens.** The TS-register module presented in Section 3.3 supports the configuration of an arbitrary number of register tokens. In Figure 8(b), we visualize the relationship between the performance on the ETT datasets under a 10% few-shot setting and the number of register tokens. It is observed that when the number of register tokens ranges from 1 to 6, the model's performance remains relatively stable, with an optimal outcome achieved when the number is set to 3. This phenomenon may be because when the number of register tokens is too small, they contain insufficient domain-specific information, which limits their effectiveness in enhancing the model's performance. Conversely, an excess of register tokens may introduce redundant information, hindering the accurate representation of domain-specific information. Additionally, we compared the results without the adjustment of a low-rank matrix on the register tokens and found that the incorporation of a low-rank matrix adjustment led to improvements across all quantities of register tokens. This finding underscores the significance of utilizing a low-rank matrix to supplement the register tokens with downstream data-specific information.

**Thresholds upper bound.** Figure 9(a) illustrates the relationship between threshold upper bound and model performance. We have observed that the upper bound of the threshold has a minimal impact on the model's performance. Generally, the information density is higher in low-frequency components compared to high-frequency ones. Therefore, the upper bound of the threshold should be biased towards the low-frequency range to balance the information content between low-frequency and high-frequency components. However, this bias should not be excessive. Our experiments indicate that an upper bound of $L/10$ performs worse than $L/5$ as an overly left-skewed threshold results in insufficient information in the low-frequency

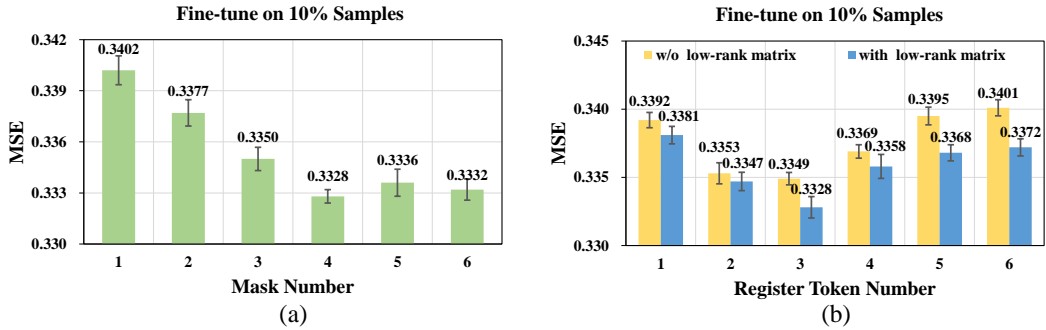

*Figure 8.* (a): Analysis of the number of masked series. (b): Analysis of the number of register tokens.

range, making the reconstruction task either too difficult or too simple. Based on our findings, we recommend using $L/5$ as the upper bound for the threshold.

**Register size.** Figure 9(b) illustrates the relationship between register size and model performance. The register size determines the upper limit of domain-specific information that the register can store. We can observe that there is a significant improvement in the model effect when the register size is increased from 32 to 128. When the register size exceeds 128, the improvement of the model effect with the increase of register size is no longer obvious. Therefore, we believe that 128 is an appropriate register size for the current pre-training datasets.

**Number of selections in Top-K strategy.** Figure 9(c) illustrates the relationship between the number of selections $k$ in Top-K strategy and model performance when we use the register to realize adaptive transfer of domain-specific information in downstream tasks. It can be seen that the model effect performance peaks at 3 tokens at $k = 3$, which has some advantages over selecting once ($k = 1$), indicating that the TopK strategy can compensate for the problem of incomplete matching of upstream and downstream domains to some extent. However, too large $k$ will also introduce redundant information and limit the accuracy of domain-specific information transfer.

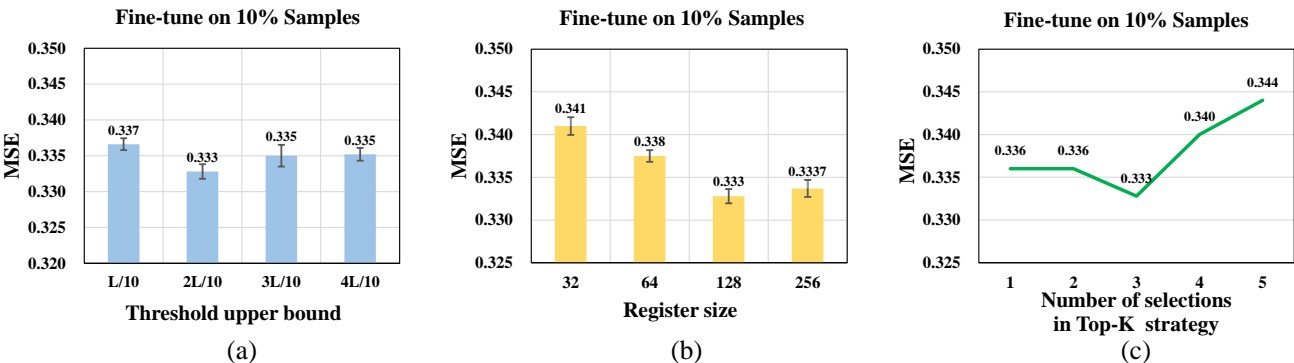

*Figure 9.* (a): Analysis of the threshold upper bound. (b): Analysis of the size of register. (c): Analysis of the number of selections in Top-K strategy.

## A.6. Short-term Forecasting

We also try to apply ROSE to short-term forecasting on the M4 (Makridakis, 2018) dataset, which contains the yearly, quarterly and monthly collected univariate marketing data. We follow TimesNet's (Wu et al., 2022) setting and metrics (SMAPE, MASE and OWA) for testing. As shown in Table 9, ROSE also exhibits competitive performance on the M4 dataset compared to the baselines.

*Table 9.* Full results on Short-term forecasting.

| Metric | ROSE | | | iTransformer | | | PatchTST | | | Timesnet | | | Dlinear | | | GPT4TS | | | S2IP-LLM | | |
|---|---|---|---|---|---|---|---|---|---|---|---|---|---|---|---|---|---|---|---|---|---|
| | SMAPE | MASE | OWA | SMAPE | MASE | OWA | SMAPE | MASE | OWA | SMAPE | MASE | OWA | SMAPE | MASE | OWA | SMAPE | MASE | OWA | SMAPE | MASE | OWA |
| Yearly | 13.302 | 3.014 | 0.833 | 13.238 | 2.952 | 0.823 | 16.766 | 4.331 | 1.018 | 13.387 | 2.996 | 0.786 | 16.965 | 4.283 | 1.058 | 13.531 | 3.015 | 0.793 | 13.413 | 3.024 | 0.792 |
| Quarterly | 9.998 | 1.165 | 0.885 | 10.001 | 1.278 | 0.949 | 12.132 | 1.513 | 0.966 | 10.100 | 1.182 | 0.890 | 12.145 | 1.520 | 1.106 | 10.100 | 1.194 | 0.898 | 10.352 | 1.228 | 0.922 |
| Monthly | 12.650 | 0.915 | 0.866 | 13.399 | 1.031 | 0.949 | 13.428 | 0.997 | 0.948 | 12.670 | 0.933 | 0.933 | 13.514 | 1.037 | 0.956 | 12.894 | 0.956 | 0.897 | 12.995 | 0.970 | 0.910 |
| Others | 4.668 | 3.126 | 1.020 | 6.558 | 4.511 | 1.401 | 6.667 | 4.834 | 1.417 | 4.891 | 3.302 | 1.035 | 6.709 | 4.953 | 1.487 | 4.940 | 3.228 | 1.029 | 4.805 | 3.247 | 1.071 |

## A.7. Model Generality

We evaluate the effectiveness of our proposed multi-frequency masking on Transformer-based models and CNN-based models, whose results are shown in Table 10. It is notable that multi-frequency masking consistently improves these forecasting models. Specifically, it achieves average improvements of 6.3%, 3.7%, 1.5% in Autoformer (Wu et al., 2021a), TimesNet (Wu et al., 2022), and PatchTST (Nie et al., 2022), respectively. This indicates that multi-frequency Masking can be widely utilized across various time series forecasting models to learn generalized time series representations and improve prediction accuracy.

*Table 10.* Performance of multi-frequency masking.

| Datasets | ETTm1 | | ETTm2 | | ETTh1 | | ETTh2 | |
|---|---|---|---|---|---|---|---|---|
| Metric | MSE | MAE | MSE | MAE | MSE | MAE | MSE | MAE |
| Autoformer | 0.600 | 0.521 | 0.328 | 0.365 | 0.493 | 0.487 | 0.452 | 0.458 |
| +Multi-frequency Masking | **0.549** | **0.488** | **0.306** | **0.349** | **0.474** | **0.478** | **0.406** | **0.425** |
| TimesNet | 0.400 | 0.406 | 0.291 | 0.333 | 0.458 | 0.450 | 0.414 | 0.427 |
| +Multi-frequency Masking | **0.386** | **0.398** | **0.282** | **0.324** | **0.446** | **0.438** | **0.386** | **0.403** |
| PatchTST | 0.353 | 0.382 | 0.256 | 0.317 | 0.413 | 0.434 | **0.331** | 0.381 |
| +Multi-frequency Masking | **0.347** | **0.372** | **0.252** | **0.308** | **0.405** | **0.424** | 0.337 | **0.379** |

## A.8. Results Deviation

We have conducted ROSE three times with different random seeds and have recorded the standard deviations for both the full-shot setting and the 10% few-shot setting, as illustrated in Table 11. As the baselines didn't report deviations in the original paper, we only reported the deviations of the PatchTST in the full-shot setting as a comparison. It can be observed that ROSE exhibits stable performance.

*Table 11.* Results deviation.

| Models | ROSE | | ROSE (10%) | | PatchTST | | confidence interval |
|---|---|---|---|---|---|---|---|
| Metric | MSE | MAE | MSE | MAE | MSE | MAE | - |
| ETTm1 | 0.342±0.003 | 0.367±0.002 | 0.349±0.003 | 0.372±0.002 | 0.349± 0.004 | 0.383±0.003 | 99% |
| ETTm2 | 0.246±0.002 | 0.303±0.004 | 0.249±0.002 | 0.308±0.002 | 0.255±0.002 | 0.314±0.003 | 99% |
| ETTh1 | 0.392±0.004 | 0.413±0.004 | 0.397±0.003 | 0.419±0.003 | 0.411±0.003 | 0.432±0.005 | 99% |
| ETTh2 | 0.330±0.003 | 0.374±0.002 | 0.335±0.004 | 0.380±0.003 | 0.348±0.004 | 0.390±0.004 | 99% |
| Traffic | 0.391±0.008 | 0.266±0.005 | 0.418±0.011 | 0.278±0.006 | 0.404±0.009 | 0.283±0.002 | 99% |
| Weather | 0.217±0.008 | 0.250±0.007 | 0.224±0.007 | 0.252±0.009 | 0.223±0.011 | 0.263±0.014 | 99% |
| Electricity | 0.156±0.007 | 0.249±0.009 | 0.164±0.004 | 0.253±0.004 | 0.163±0.009 | 0.261±0.013 | 99% |

## A.9. Visualization

### A.9.1. Visualization analysis

To showcase the benefits of cross-domain pre-training, we performed visualizations in both the zero-shot setting and full-shot setting.

**Zero-shot**: We pre-train the baselines iTransformer and PatchTST on the energy domain dataset ETTm1 and test their zero-shot performance on two different domains (weather, traffic) . ROSE, without fine-tuning, is evaluated to the same two test-sets. As shown in the Figure 10, we find that the baselines generally perform worse during domain shift due to their poor generalization. However, ROSE excels in scenarios across all domains, which demonstrates the benefits of cross-domain pre-training for improving generalization.

**Full-shot**: We train the baselines on the train-set of downstream dataset ETTh2 and fine-tune ROSE on the same train-set. As shown in the Figure 11, We find that the baselines is limited by data diversity, leading to poor performance on patterns which rarely appear. However, ROSE excels in these cases, as the cross-domain pre-training allows ROSE to learn diverse temporal patterns, and helps ROSE to predict the patterns which rarely appear in the downstream train-set well.

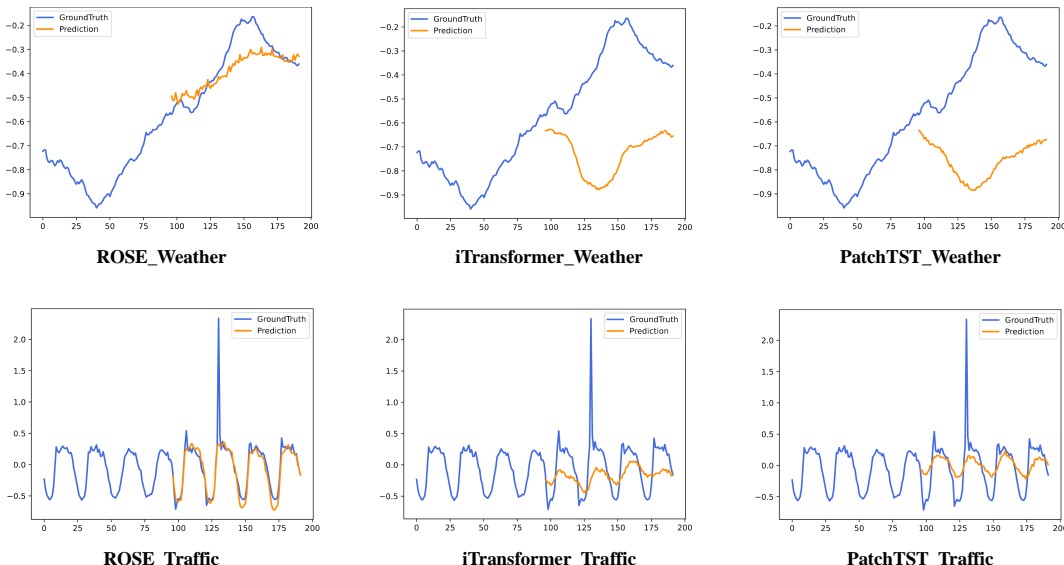

*Figure 10.* Visualization comparison of ROSE with cross-domain pre-training and other SOTA baselines in the zero-shot setting for three domain datasets.

### A.9.2. Visualization showcase

To provide a distinct comparison among different models, we present visualizations of the forecasting results on the ETTh2 dataset and the weather dataset in different settings, as shown in Figures 12 to Figures 15, given by the following models: DLinear (Zeng et al., 2023), TimesNet (Wu et al., 2022), iTransfomrer (Liu et al., 2023), and PatchTST (Nie et al., 2022). Among the methods, ROSE demonstrates the most accurate prediction ability.

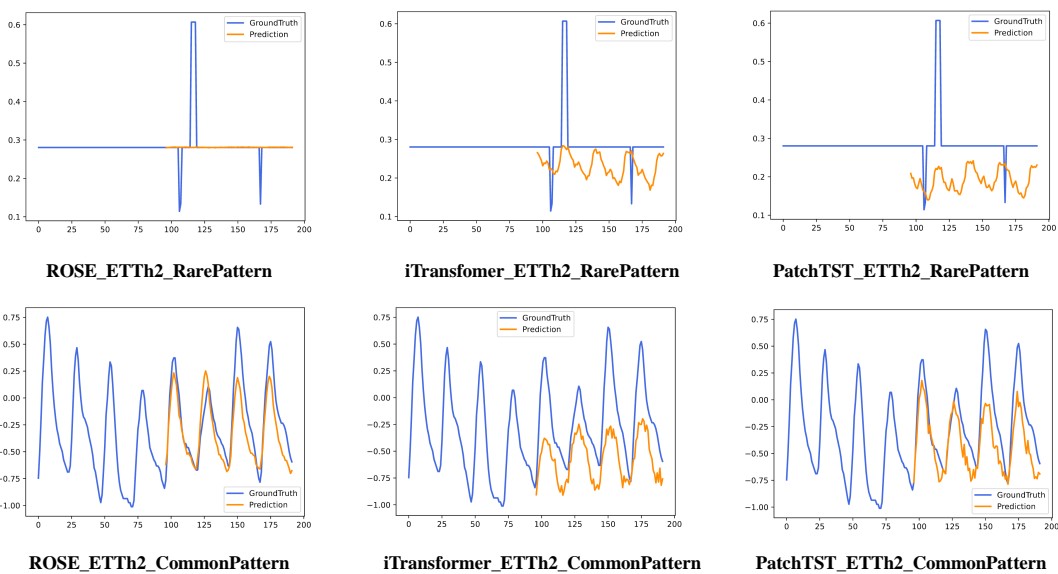

*Figure 11.* Visualization comparison of ROSE with cross-domain pre-training and other SOTA baselines in the full-shot setting for rare and common patterns.

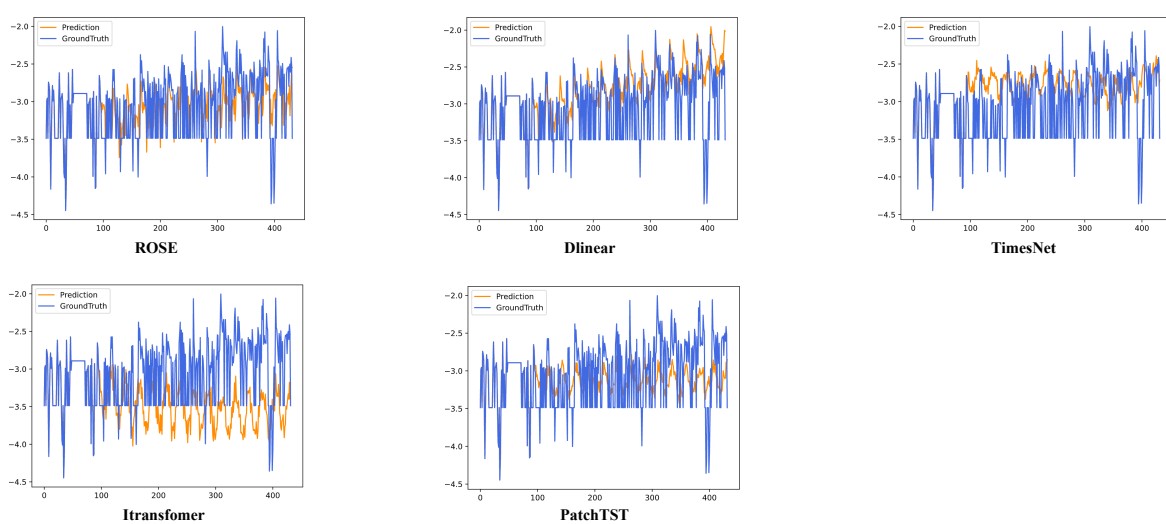

*Figure 12.* Visualization of input-512 and predict-336 forecasting results on the ETTh2 dataset in full-shot setting.

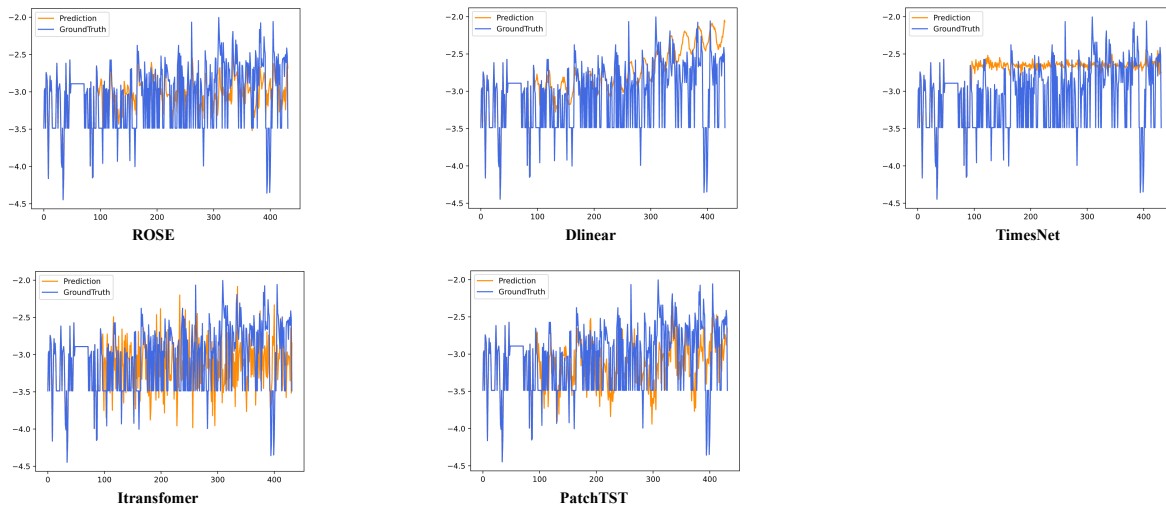

*Figure 13.* Visualization of input-512 and predict-336 forecasting results on the ETTh2 dataset in 10% few-shot setting.

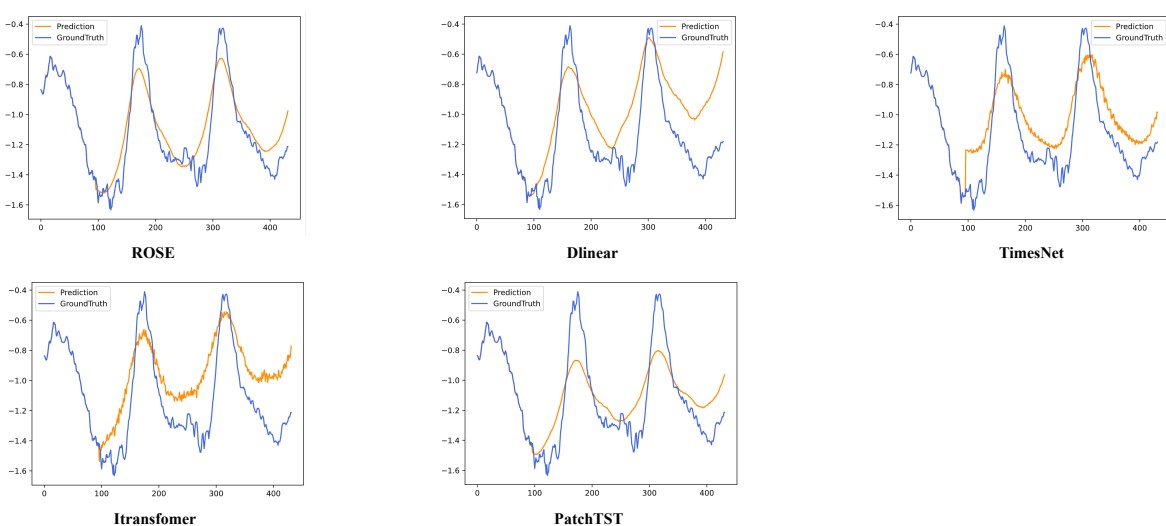

*Figure 14.* Visualization of input-512 and predict-336 forecasting results on the weather dataset in full-shot setting.

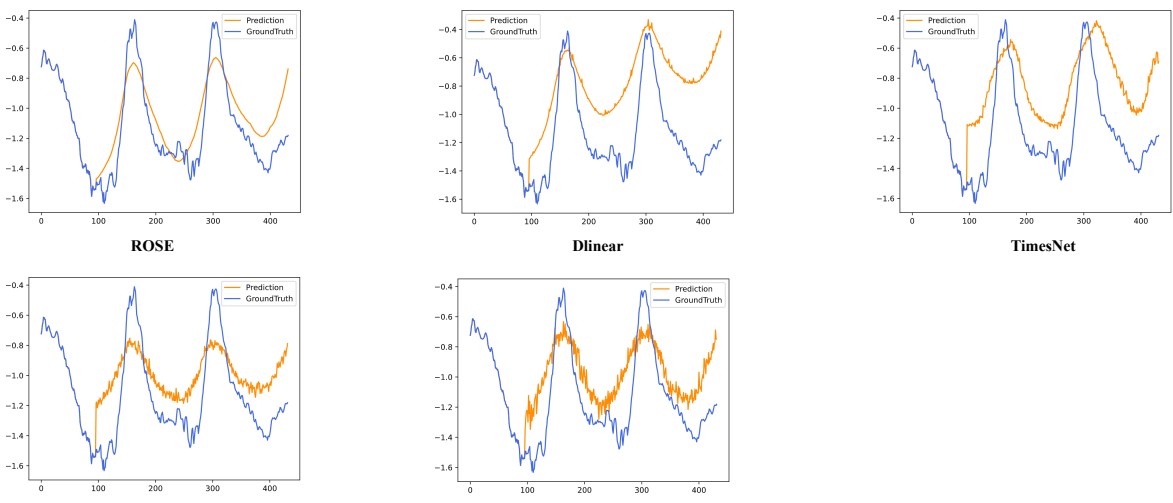

*Figure 15.* Visualization of input-512 and predict-336 forecasting results on the weather dataset in 10% few-shot setting.

## A.10. Full Results

### A.10.1. FULL-SHOT RESULTS

Table 12 shows the full results of ROSE in full-shot setting for four prediction lengths. ROSE shows the advantage over the specific models and LLM-based models trained with the full training set.

*Table 12.* Full results in full-shot setting.

| Models | | ReadyTS | | ITransformer | | PatchTST | | Timesnet | | Dlinear | | GPT4TS | | S²IP-LLM | |
|---|---|---|---|---|---|---|---|---|---|---|---|---|---|---|---|
| Metric | | MSE | MAE | MSE | MAE | MSE | MAE | MSE | MAE | MSE | MAE | MSE | MAE | MSE | MAE |
| ETTh1 | 96 | **0.354** | **0.385** | 0.386 | 0.405 | 0.370 | 0.400 | 0.470 | 0.470 | 0.367 | 0.396 | 0.376 | 0.397 | 0.366 | 0.396 |
| | 192 | **0.389** | **0.407** | 0.424 | 0.440 | 0.413 | 0.429 | 0.568 | 0.523 | 0.400 | 0.417 | 0.416 | 0.418 | 0.401 | 0.420 |
| | 336 | **0.406** | **0.422** | 0.449 | 0.460 | 0.422 | 0.440 | 0.595 | 0.547 | 0.428 | 0.439 | 0.442 | 0.433 | 0.412 | 0.431 |
| | 720 | **0.413** | **0.443** | 0.495 | 0.487 | 0.447 | 0.468 | 0.694 | 0.591 | 0.468 | 0.491 | 0.477 | 0.456 | 0.440 | 0.458 |
| | avg | **0.391** | **0.414** | 0.439 | 0.448 | 0.413 | 0.434 | 0.582 | 0.533 | 0.416 | 0.436 | 0.427 | 0.426 | 0.406 | 0.427 |
| ETTh2 | 96 | **0.265** | **0.320** | 0.297 | 0.348 | 0.274 | 0.337 | 0.351 | 0.399 | 0.302 | 0.368 | 0.285 | 0.342 | 0.278 | 0.340 |
| | 192 | 0.328 | **0.369** | 0.371 | 0.403 | 0.341 | 0.382 | 0.394 | 0.429 | 0.404 | 0.433 | 0.354 | 0.389 | **0.246** | 0.385 |
| | 336 | 0.353 | 0.391 | 0.404 | 0.428 | **0.329** | **0.384** | 0.415 | 0.443 | 0.511 | 0.498 | 0.373 | 0.407 | 0.367 | 0.406 |
| | 720 | **0.376** | **0.417** | 0.424 | 0.444 | 0.379 | 0.422 | 0.477 | 0.481 | 0.815 | 0.640 | 0.406 | 0.441 | 0.400 | 0.436 |
| | avg | **0.331** | **0.374** | 0.374 | 0.406 | **0.331** | 0.381 | 0.409 | 0.438 | 0.508 | 0.485 | 0.354 | 0.394 | 0.347 | 0.391 |
| ETTm1 | 96 | **0.275** | 0.328 | 0.300 | 0.353 | 0.293 | 0.346 | 0.405 | 0.421 | 0.303 | 0.346 | 0.292 | **0.262** | 0.288 | 0.346 |
| | 192 | 0.324 | 0.358 | 0.345 | 0.382 | 0.333 | 0.370 | 0.508 | 0.473 | 0.335 | 0.365 | 0.332 | **0.301** | 0.323 | 0.365 |
| | 336 | 0.354 | 0.377 | 0.374 | 0.398 | 0.369 | 0.392 | 0.523 | 0.479 | 0.365 | 0.384 | 0.366 | **0.341** | 0.359 | 0.390 |
| | 720 | 0.411 | 0.407 | 0.429 | 0.430 | 0.416 | 0.420 | 0.523 | 0.484 | 0.418 | 0.415 | 0.417 | **0.401** | **0.403** | 0.418 |
| | avg | **0.341** | **0.367** | 0.362 | 0.391 | 0.353 | 0.382 | 0.490 | 0.464 | 0.356 | 0.378 | 0.352 | 0.383 | 0.343 | 0.379 |
| ETTm2 | 96 | **0.157** | **0.243** | 0.175 | 0.266 | 0.166 | 0.256 | 0.233 | 0.305 | 0.164 | 0.255 | 0.173 | 0.262 | 0.165 | 0.257 |
| | 192 | **0.213** | **0.283** | 0.242 | 0.312 | 0.223 | 0.296 | 0.265 | 0.328 | 0.224 | 0.304 | 0.229 | 0.301 | 0.222 | 0.299 |
| | 336 | **0.266** | **0.319** | 0.282 | 0.340 | 0.274 | 0.329 | 0.379 | 0.392 | 0.277 | 0.339 | 0.286 | 0.341 | 0.277 | 0.330 |
| | 720 | **0.347** | **0.373** | 0.378 | 0.398 | 0.362 | 0.385 | 0.390 | 0.407 | 0.371 | 0.401 | 0.378 | 0.401 | 0.363 | 0.390 |
| | avg | **0.246** | **0.305** | 0.269 | 0.329 | 0.256 | 0.317 | 0.317 | 0.358 | 0.259 | 0.325 | 0.266 | 0.326 | 0.257 | 0.319 |
| Weather | 96 | **0.145** | **0.182** | 0.159 | 0.208 | 0.149 | 0.198 | 0.193 | 0.244 | 0.170 | 0.230 | 0.162 | 0.212 | **0.145** | 0.195 |
| | 192 | **0.183** | **0.226** | 0.200 | 0.248 | 0.194 | 0.241 | 0.320 | 0.329 | 0.212 | 0.267 | 0.204 | 0.248 | 0.190 | 0.235 |
| | 336 | **0.232** | **0.267** | 0.253 | 0.289 | 0.245 | 0.282 | 0.363 | 0.366 | 0.257 | 0.305 | 0.254 | 0.286 | 0.243 | 0.280 |
| | 720 | **0.309** | 0.327 | 0.321 | 0.338 | 0.314 | 0.334 | 0.440 | 0.404 | 0.318 | 0.356 | 0.326 | 0.337 | 0.312 | **0.326** |
| | avg | **0.217** | **0.251** | 0.233 | 0.271 | 0.226 | 0.264 | 0.329 | 0.336 | 0.239 | 0.289 | 0.237 | 0.270 | 0.222 | 0.259 |
| Electricity | 96 | **0.125** | **0.220** | 0.138 | 0.237 | 0.129 | 0.222 | 0.182 | 0.287 | 0.141 | 0.241 | 0.139 | 0.238 | 0.135 | 0.230 |
| | 192 | **0.142** | **0.235** | 0.157 | 0.256 | 0.147 | 0.240 | 0.193 | 0.293 | 0.154 | 0.254 | 0.153 | 0.251 | 0.149 | 0.247 |
| | 336 | **0.162** | **0.252** | 0.167 | 0.264 | 0.163 | 0.259 | 0.196 | 0.298 | 0.168 | 0.271 | 0.169 | 0.266 | 0.167 | 0.266 |
| | 720 | **0.191** | **0.284** | 0.194 | 0.286 | 0.197 | 0.290 | 0.209 | 0.307 | 0.203 | 0.303 | 0.206 | 0.297 | 0.200 | 0.287 |
| | avg | **0.155** | **0.248** | 0.164 | 0.261 | 0.159 | 0.253 | 0.195 | 0.296 | 0.166 | 0.267 | 0.167 | 0.263 | 0.161 | 0.257 |
| Traffic | 96 | **0.354** | 0.252 | 0.363 | 0.265 | 0.360 | **0.249** | 0.611 | 0.323 | 0.411 | 0.294 | 0.388 | 0.282 | 0.379 | 0.274 |
| | 192 | **0.377** | 0.257 | 0.385 | 0.273 | 0.379 | **0.256** | 0.609 | 0.327 | 0.421 | 0.298 | 0.407 | 0.290 | 0.397 | 0.282 |
| | 336 | 0.396 | **0.262** | 0.396 | 0.277 | **0.392** | 0.264 | 0.616 | 0.335 | 0.431 | 0.304 | 0.412 | 0.294 | 0.407 | 0.289 |
| | 720 | 0.434 | **0.283** | 0.445 | 0.312 | **0.432** | 0.286 | 0.656 | 0.349 | 0.468 | 0.325 | 0.450 | 0.312 | 0.440 | 0.301 |
| | avg | **0.390** | **0.264** | 0.397 | 0.282 | 0.391 | **0.264** | 0.623 | 0.333 | 0.433 | 0.305 | 0.414 | 0.294 | 0.405 | 0.286 |

A.10.2. FEW-SHOT RESULTS

Table 13 shows the full results of ROSE in 10% few-shot setting for four prediction lengths. ROSE shows the advantage over the specific models and LLM-based models trained with the 10% training set.

Table 13. Full results in 10% few-shot setting

| Models | | ReadyTS | | ITransformer | | PatchTST | | Timesnet | | Dlinear | | GPT4TS | | S$^2$IP-LLM | |
|---|---|---|---|---|---|---|---|---|---|---|---|---|---|---|---|
| Metric | | MSE | MAE | MSE | MAE | MSE | MAE | MSE | MAE | MSE | MAE | MSE | MAE | MSE | MAE |
| ETTh1 | 96 | **0.367** | **0.395** | 0.442 | 0.464 | 0.458 | 0.463 | 0.579 | 0.522 | 1.355 | 0.816 | 0.458 | 0.456 | 0.481 | 0.474 |
| | 192 | **0.399** | **0.416** | 0.476 | 0.475 | 0.481 | 0.490 | 0.641 | 0.553 | 1.210 | 0.825 | 0.570 | 0.516 | 0.518 | 0.491 |
| | 336 | **0.405** | **0.423** | 0.486 | 0.482 | 0.465 | 0.475 | 0.721 | 0.582 | 1.487 | 0.914 | 0.608 | 0.535 | 0.664 | 0.570 |
| | 720 | **0.416** | **0.443** | 0.509 | 0.506 | 0.478 | 0.492 | 0.630 | 0.574 | 1.369 | 0.826 | 0.725 | 0.591 | 0.711 | 0.584 |
| | avg | **0.397** | **0.419** | 0.478 | 0.482 | 0.470 | 0.480 | 0.643 | 0.558 | 1.355 | 0.845 | 0.590 | 0.525 | 0.593 | 0.529 |
| ETTh2 | 96 | **0.273** | **0.332** | 0.333 | 0.385 | 0.350 | 0.389 | 0.378 | 0.413 | 1.628 | 0.724 | 0.331 | 0.374 | 0.354 | 0.400 |
| | 192 | **0.334** | **0.376** | 0.402 | 0.428 | 0.416 | 0.426 | 0.463 | 0.460 | 1.388 | 0.713 | 0.402 | 0.411 | 0.400 | 0.423 |
| | 336 | **0.358** | **0.397** | 0.438 | 0.452 | 0.401 | 0.429 | 0.507 | 0.495 | 1.595 | 0.772 | 0.406 | 0.433 | 0.442 | 0.450 |
| | 720 | **0.376** | **0.417** | 0.466 | 0.477 | 0.436 | 0.457 | 0.516 | 0.501 | 1.664 | 0.857 | 0.449 | 0.464 | 0.480 | 0.486 |
| | avg | **0.335** | **0.380** | 0.410 | 0.436 | 0.401 | 0.425 | 0.466 | 0.467 | 1.569 | 0.766 | 0.397 | 0.421 | 0.419 | 0.439 |
| ETTm1 | 96 | **0.287** | **0.336** | 0.353 | 0.392 | 0.317 | 0.363 | 0.481 | 0.446 | 0.454 | 0.475 | 0.390 | 0.404 | 0.388 | 0.401 |
| | 192 | **0.331** | **0.362** | 0.385 | 0.410 | 0.351 | 0.382 | 0.621 | 0.491 | 0.575 | 0.548 | 0.429 | 0.423 | 0.422 | 0.421 |
| | 336 | **0.362** | **0.379** | 0.422 | 0.432 | 0.376 | 0.398 | 0.521 | 0.479 | 0.773 | 0.631 | 0.469 | 0.439 | 0.456 | 0.430 |
| | 720 | **0.416** | **0.412** | 0.494 | 0.472 | 0.435 | 0.430 | 0.571 | 0.508 | 0.943 | 0.716 | 0.569 | 0.498 | 0.554 | 0.490 |
| | avg | **0.349** | **0.372** | 0.414 | 0.426 | 0.370 | 0.393 | 0.549 | 0.481 | 0.686 | 0.593 | 0.464 | 0.441 | 0.455 | 0.435 |
| ETTm2 | 96 | **0.159** | **0.247** | 0.183 | 0.279 | 0.170 | 0.259 | 0.212 | 0.292 | 0.493 | 0.476 | 0.188 | 0.269 | 0.192 | 0.274 |
| | 192 | **0.217** | **0.287** | 0.247 | 0.320 | 0.226 | 0.297 | 0.297 | 0.353 | 0.923 | 0.658 | 0.251 | 0.309 | 0.246 | 0.313 |
| | 336 | **0.269** | **0.322** | 0.300 | 0.353 | 0.284 | 0.333 | 0.328 | 0.364 | 1.407 | 0.822 | 0.307 | 0.346 | 0.301 | 0.340 |
| | 720 | **0.357** | **0.377** | 0.385 | 0.408 | 0.363 | 0.382 | 0.456 | 0.440 | 1.626 | 0.905 | 0.426 | 0.417 | 0.400 | 0.403 |
| | avg | **0.250** | **0.308** | 0.279 | 0.340 | 0.261 | 0.318 | 0.323 | 0.362 | 1.112 | 0.715 | 0.293 | 0.335 | 0.284 | 0.332 |
| Weather | 96 | **0.145** | **0.184** | 0.189 | 0.229 | 0.166 | 0.217 | 0.199 | 0.248 | 0.230 | 0.318 | 0.163 | 0.215 | 0.159 | 0.210 |
| | 192 | **0.190** | **0.227** | 0.239 | 0.269 | 0.211 | 0.257 | 0.249 | 0.285 | 0.357 | 0.425 | 0.210 | 0.254 | 0.200 | 0.251 |
| | 336 | **0.245** | **0.269** | 0.294 | 0.308 | 0.261 | 0.296 | 0.297 | 0.316 | 0.464 | 0.493 | 0.256 | 0.292 | 0.257 | 0.293 |
| | 720 | **0.317** | **0.328** | 0.366 | 0.356 | 0.328 | 0.342 | 0.367 | 0.361 | 0.515 | 0.532 | 0.321 | 0.339 | 0.317 | 0.335 |
| | avg | **0.224** | **0.252** | 0.272 | 0.291 | 0.242 | 0.278 | 0.278 | 0.303 | 0.391 | 0.442 | 0.238 | 0.275 | 0.233 | 0.272 |
| Electricity | 96 | **0.135** | **0.226** | 0.184 | 0.276 | 0.161 | 0.256 | 0.279 | 0.359 | 0.227 | 0.334 | 0.139 | 0.237 | 0.143 | 0.243 |
| | 192 | **0.150** | **0.240** | 0.192 | 0.284 | 0.163 | 0.257 | 0.282 | 0.363 | 0.265 | 0.366 | 0.156 | 0.252 | 0.159 | 0.258 |
| | 336 | **0.166** | **0.258** | 0.216 | 0.308 | 0.173 | 0.266 | 0.289 | 0.367 | 0.339 | 0.417 | 0.175 | 0.270 | 0.170 | 0.269 |
| | 720 | **0.205** | **0.290** | 0.265 | 0.347 | 0.221 | 0.313 | 0.333 | 0.399 | 0.482 | 0.478 | 0.233 | 0.317 | 0.230 | 0.315 |
| | avg | **0.164** | **0.253** | 0.214 | 0.304 | 0.180 | 0.273 | 0.296 | 0.372 | 0.328 | 0.399 | 0.176 | 0.269 | 0.175 | 0.271 |
| Traffic | 96 | **0.398** | **0.270** | 0.458 | 0.314 | 0.421 | 0.299 | 0.705 | 0.386 | 0.616 | 0.385 | 0.414 | 0.297 | 0.403 | 0.293 |
| | 192 | **0.405** | **0.270** | 0.473 | 0.319 | 0.439 | 0.313 | 0.710 | 0.393 | 0.710 | 0.480 | 0.426 | 0.301 | 0.412 | 0.295 |
| | 336 | **0.417** | **0.277** | 0.491 | 0.329 | 0.448 | 0.318 | 0.863 | 0.456 | 0.723 | 0.481 | 0.434 | 0.303 | 0.427 | 0.316 |
| | 720 | **0.452** | **0.294** | 0.536 | 0.361 | 0.478 | 0.320 | 0.928 | 0.485 | 0.673 | 0.436 | 0.487 | 0.337 | 0.469 | 0.325 |
| | avg | **0.418** | **0.278** | 0.490 | 0.331 | 0.447 | 0.312 | 0.801 | 0.430 | 0.680 | 0.446 | 0.440 | 0.310 | 0.427 | 0.307 |

### A.10.3. ABLATION STUDY RESULTS

**Novelty of decomposed frequency learning.** Frequency masking is not a new concept, but past approaches randomly mask frequencies of a single time series once (Chen et al., 2023b; Zhang et al., 2023), which show limited forecasting effectiveness due to the lack of common pattern learning from heterogeneous time series that come from various domains. While the multi-frequency masking we proposed randomly mask either high-frequency or low-frequency components of a time series multiple times as the key to enable learning of common time series patterns, such as trends and various long and short term fluctuations. Moreover, different from utilizing frequency masking as a way of data augmentation to enhance the diversity of input data (Chen et al., 2023b; Zhang et al., 2023), we combine multi-frequency masking with reconstruction task as a novel pre-training framework, that learns a universal and unified feature representation by comprehending the data from various frequency perspectives, thereby enabling it to learn generalized representations.

**Difference between frequency-domain masking and time-domain noise addition.** Multi-frequency masking and reconstruction are not equivalent to the pre-training methods of adding noise and denoising (Noise). Due to the sparsity of time series, the process of adding noise and denoising may potentially disrupt the information of original time series (Dong et al., 2024). In contrast, multi-frequency masking not only preserves the series from such disruption but also helps the model understand temporal patterns from a multi-frequency perspective, thereby helping the model to learn general features better.

**Other pre-training tasks.** Based on the two points above, we conduct experiments to compare two other pre-training tasks: 1) using frequency-domain augmentation only for data expansion without reconstruction task (*Aug*); 2) replacing multi-frequency masking and reconstruction task with adding time-domian noise and denoise task (*Noise*).As shown in Table 14, we find that ROSE is significantly more effective than *Aug* and *Noise*, which demonstrates the effectiveness of multi-frequency masking and reconstruction task in learning generalized features.

### A.10.4. ZERO-SHOT RESULTS

Table 15 shows the full results of ROSE and other foundation models in zero-shot setting for four prediction lengths. ROSE exhibits competitive performance.

*Table 14.* Full results of ablation study

| Design | | Pred_len | ETTm1 | | ETTm2 | | ETTh1 | | ETTh2 | |
|---|---|---|---|---|---|---|---|---|---|---|
| | | | MSE | MAE | MSE | MAE | MSE | MAE | MSE | MAE |
| ReadyTS | | 96 | **0.287** | **0.336** | **0.159** | **0.247** | **0.367** | **0.395** | **0.273** | **0.332** |
| | | 192 | **0.331** | **0.362** | **0.217** | **0.287** | **0.399** | **0.416** | **0.334** | **0.376** |
| | | 336 | 0.362 | **0.379** | **0.269** | **0.322** | **0.405** | **0.423** | **0.358** | **0.397** |
| | | 720 | **0.416** | **0.412** | **0.357** | **0.377** | **0.416** | **0.443** | **0.376** | **0.417** |
| | | avg | **0.349** | **0.372** | **0.250** | **0.308** | **0.397** | **0.419** | **0.335** | **0.380** |
| Replace Multi-Frequency Masking | Random Frequency Masking | 96 | 0.330 | 0.370 | 0.170 | 0.262 | 0.391 | 0.399 | 0.303 | 0.359 |
| | | 192 | 0.352 | 0.392 | 0.232 | 0.298 | 0.406 | 0.430 | 0.356 | 0.395 |
| | | 336 | 0.390 | 0.389 | 0.276 | 0.342 | 0.411 | 0.432 | 0.417 | 0.428 |
| | | 720 | 0.452 | 0.438 | 0.366 | 0.392 | 0.432 | 0.447 | 0.420 | 0.439 |
| | | avg | 0.381 | 0.397 | 0.261 | 0.324 | 0.410 | 0.427 | 0.374 | 0.405 |
| | Multi-Patch Masking | 96 | 0.302 | 0.348 | 0.168 | 0.257 | 0.377 | 0.408 | 0.282 | 0.343 |
| | | 192 | 0.336 | 0.367 | 0.228 | 0.297 | 0.404 | 0.423 | 0.343 | 0.379 |
| | | 336 | 0.364 | 0.385 | 0.277 | 0.328 | 0.405 | 0.420 | 0.374 | 0.403 |
| | | 720 | 0.423 | 0.416 | 0.364 | 0.381 | 0.431 | 0.455 | 0.396 | 0.430 |
| | | avg | 0.356 | 0.379 | 0.259 | 0.316 | 0.404 | 0.426 | 0.349 | 0.389 |
| | Patch Masking | 96 | 0.318 | 0.366 | 0.168 | 0.259 | 0.388 | 0.412 | 0.303 | 0.359 |
| | | 192 | 0.355 | 0.388 | 0.228 | 0.298 | 0.402 | 0.422 | 0.370 | 0.399 |
| | | 336 | 0.388 | 0.406 | 0.279 | 0.331 | 0.411 | 0.435 | 0.413 | 0.428 |
| | | 720 | 0.450 | 0.438 | 0.370 | 0.388 | 0.431 | 0.459 | 0.413 | 0.443 |
| | | avg | 0.378 | 0.400 | 0.261 | 0.319 | 0.408 | 0.432 | 0.375 | 0.407 |
| Other pre-training tasks | Aug | 96 | 0.304 | 0.357 | 0.178 | 0.266 | 0.376 | 0.405 | 0.281 | 0.348 |
| | | 192 | 0.343 | 0.379 | 0.254 | 0.318 | 0.409 | 0.429 | 0.345 | 0.389 |
| | | 336 | 0.373 | 0.400 | 0.299 | 0.354 | 0.435 | 0.453 | 0.382 | 0.417 |
| | | 720 | 0.444 | 0.432 | 0.387 | 0.408 | 0.452 | 0.471 | 0.434 | 0.452 |
| | | avg | 0.366 | 0.392 | 0.279 | 0.336 | 0.418 | 0.439 | 0.360 | 0.401 |
| | Noise | 96 | 0.303 | 0.355 | 0.172 | 0.261 | 0.370 | 0.405 | 0.280 | 0.341 |
| | | 192 | 0.342 | 0.376 | 0.221 | 0.292 | 0.403 | 0.427 | 0.350 | 0.384 |
| | | 336 | 0.368 | 0.393 | 0.272 | 0.325 | 0.420 | 0.439 | 0.385 | 0.410 |
| | | 720 | 0.423 | 0.422 | 0.367 | 0.386 | 0.442 | 0.462 | 0.403 | 0.432 |
| | | avg | 0.359 | 0.387 | 0.258 | 0.316 | 0.409 | 0.433 | 0.355 | 0.392 |
| w/o | TS-Register | 96 | 0.297 | 0.345 | 0.164 | 0.252 | 0.379 | 0.399 | 0.276 | 0.336 |
| | | 192 | 0.334 | 0.367 | 0.221 | 0.290 | 0.419 | 0.420 | 0.350 | 0.380 |
| | | 336 | **0.360** | 0.384 | 0.275 | 0.325 | 0.438 | 0.442 | 0.393 | 0.411 |
| | | 720 | 0.424 | 0.416 | 0.364 | 0.379 | 0.435 | 0.448 | 0.400 | 0.432 |
| | | avg | 0.354 | 0.378 | 0.256 | 0.312 | 0.418 | 0.427 | 0.355 | 0.390 |
| | Prediction Task | 96 | 0.301 | 0.348 | 0.166 | 0.255 | 0.380 | 0.407 | 0.295 | 0.359 |
| | | 192 | 0.343 | 0.374 | 0.221 | 0.291 | 0.410 | 0.426 | 0.372 | 0.406 |
| | | 336 | 0.374 | 0.393 | 0.275 | 0.327 | 0.440 | 0.443 | 0.403 | 0.429 |
| | | 720 | 0.424 | 0.420 | 0.366 | 0.384 | 0.458 | 0.476 | 0.418 | 0.446 |
| | | avg | 0.360 | 0.384 | 0.257 | 0.314 | 0.422 | 0.438 | 0.372 | 0.410 |
| | Reconstruction Task | 96 | 0.329 | 0.371 | 0.175 | 0.265 | 0.374 | 0.399 | 0.296 | 0.355 |
| | | 192 | 0.363 | 0.391 | 0.233 | 0.304 | 0.407 | 0.422 | 0.354 | 0.389 |
| | | 336 | 0.394 | 0.407 | 0.287 | 0.340 | 0.437 | 0.440 | 0.385 | 0.413 |
| | | 720 | 0.461 | 0.442 | 0.379 | 0.396 | 0.430 | 0.453 | 0.408 | 0.438 |
| | | avg | 0.387 | 0.403 | 0.269 | 0.327 | 0.412 | 0.428 | 0.361 | 0.399 |
| From Scratch | | 96 | 0.301 | 0.357 | 0.171 | 0.260 | 0.419 | 0.439 | 0.315 | 0.389 |
| | | 192 | 0.358 | 0.385 | 0.223 | 0.294 | 0.438 | 0.457 | 0.366 | 0.405 |
| | | 336 | 0.390 | 0.396 | 0.282 | 0.336 | 0.484 | 0.484 | 0.424 | 0.435 |
| | | 720 | 0.436 | 0.427 | 0.366 | 0.380 | 0.540 | 0.538 | 0.495 | 0.473 |
| | | avg | 0.371 | 0.391 | 0.261 | 0.318 | 0.470 | 0.480 | 0.400 | 0.425 |

Table 15. Full results in zero-shot setting.

| Models | | ROSE_512 | | Timer | | MOIRAI | | Chronos | | TimesFM | | Moment | |
|---|---|---|---|---|---|---|---|---|---|---|---|---|---|
| Metric | | MSE | MAE | MSE | MAE | MSE | MAE | MSE | MAE | MSE | MAE | MSE | MAE |
| ETTh1 | 96 | **0.382** | 0.408 | 0.414 | 0.439 | 0.405 | **0.397** | 0.494 | 0.409 | 0.432 | 0.405 | 0.706 | 0.561 |
| | 192 | **0.400** | **0.420** | 0.440 | 0.455 | 0.458 | 0.428 | 0.561 | 0.443 | 0.492 | 0.438 | 0.716 | 0.579 |
| | 336 | **0.404** | **0.426** | 0.455 | 0.463 | 0.509 | 0.454 | 0.580 | 0.460 | 0.519 | 0.458 | 0.705 | 0.583 |
| | 720 | **0.420** | **0.447** | 0.496 | 0.496 | 0.529 | 0.494 | 0.605 | 0.495 | 0.512 | 0.477 | 0.705 | 0.597 |
| | avg | **0.401** | **0.425** | 0.451 | 0.463 | 0.475 | 0.443 | 0.560 | 0.452 | 0.489 | 0.444 | 0.708 | 0.580 |
| ETTh2 | 96 | **0.298** | 0.362 | 0.305 | 0.355 | 0.303 | **0.338** | 0.306 | **0.338** | 0.311 | 0.345 | 0.373 | 0.416 |
| | 192 | **0.336** | 0.385 | 0.365 | 0.406 | 0.369 | **0.384** | 0.396 | 0.394 | 0.401 | 0.397 | 0.384 | 0.422 |
| | 336 | **0.353** | **0.399** | 0.378 | 0.413 | 0.397 | 0.410 | 0.423 | 0.417 | 0.436 | 0.430 | 0.386 | 0.426 |
| | 720 | **0.395** | **0.432** | 0.414 | 0.457 | 0.447 | 0.450 | 0.442 | 0.439 | 0.437 | 0.450 | 0.425 | 0.454 |
| | avg | **0.346** | **0.394** | 0.366 | 0.408 | 0.379 | 0.396 | 0.392 | 0.397 | 0.396 | 0.405 | 0.392 | 0.430 |
| ETTm1 | 96 | 0.512 | 0.460 | 0.440 | 0.422 | 0.660 | 0.476 | 0.514 | 0.443 | **0.366** | **0.374** | 0.679 | 0.544 |
| | 192 | 0.512 | 0.462 | 0.505 | 0.458 | 0.707 | 0.500 | 0.608 | 0.475 | **0.413** | **0.401** | 0.690 | 0.550 |
| | 336 | 0.523 | 0.470 | 0.570 | 0.490 | 0.730 | 0.515 | 0.690 | 0.507 | **0.445** | **0.429** | 0.701 | 0.557 |
| | 720 | 0.552 | 0.490 | 0.659 | 0.534 | 0.758 | 0.536 | 0.733 | 0.555 | **0.513** | **0.470** | 0.719 | 0.569 |
| | avg | 0.525 | 0.471 | 0.544 | 0.476 | 0.714 | 0.507 | 0.636 | 0.495 | **0.434** | **0.419** | 0.697 | 0.555 |
| ETTm2 | 96 | 0.224 | 0.309 | 0.203 | 0.285 | 0.216 | 0.282 | 0.202 | 0.293 | **0.189** | **0.257** | 0.230 | 0.308 |
| | 192 | 0.266 | 0.333 | **0.265** | 0.327 | 0.294 | 0.330 | 0.286 | 0.348 | 0.277 | **0.325** | 0.285 | 0.338 |
| | 336 | **0.310** | **0.358** | 0.319 | 0.361 | 0.368 | 0.373 | 0.355 | 0.386 | 0.350 | 0.381 | 0.339 | 0.369 |
| | 720 | **0.395** | **0.407** | 0.405 | 0.410 | 0.494 | 0.439 | 0.409 | 0.425 | 0.464 | 0.448 | 0.423 | 0.424 |
| | avg | **0.299** | **0.352** | 0.360 | 0.386 | 0.343 | 0.356 | 0.313 | 0.363 | 0.320 | 0.353 | 0.319 | 0.360 |
| Weather | 96 | **0.200** | 0.260 | 0.190 | **0.236** | 0.188 | 0.250 | 0.209 | 0.244 | - | - | 0.216 | 0.271 |
| | 192 | 0.239 | 0.288 | 0.261 | 0.293 | **0.237** | **0.284** | 0.254 | 0.288 | - | - | 0.264 | 0.306 |
| | 336 | **0.279** | **0.315** | 0.332 | 0.340 | 0.282 | 0.323 | 0.301 | 0.332 | - | - | 0.313 | 0.336 |
| | 720 | **0.340** | 0.357 | 0.385 | 0.381 | 0.359 | **0.345** | 0.388 | 0.374 | - | - | 0.369 | 0.380 |
| | avg | **0.265** | 0.305 | 0.292 | 0.312 | 0.267 | **0.300** | 0.288 | 0.310 | - | - | 0.291 | 0.323 |
| Electricity | 96 | 0.209 | 0.307 | 0.210 | 0.312 | 0.212 | 0.301 | **0.194** | **0.266** | - | - | 0.844 | 0.761 |
| | 192 | 0.219 | 0.315 | 0.239 | 0.337 | 0.225 | 0.320 | **0.218** | **0.289** | - | - | 0.850 | 0.762 |
| | 336 | 0.236 | 0.330 | 0.284 | 0.372 | 0.245 | 0.333 | 0.244 | **0.321** | - | - | 0.862 | 0.766 |
| | 720 | **0.273** | **0.328** | 0.456 | 0.479 | 0.282 | 0.358 | 0.324 | 0.371 | - | - | 0.888 | 0.774 |
| | avg | **0.234** | 0.320 | 0.297 | 0.375 | 0.241 | 0.328 | 0.245 | **0.312** | - | - | 0.861 | 0.766 |
| Traffic | 96 | 0.572 | 0.407 | **0.526** | **0.368** | - | - | 0.562 | 0.378 | - | - | 1.390 | 0.800 |
| | 192 | 0.575 | 0.406 | **0.561** | **0.385** | - | - | 0.579 | 0.412 | - | - | 1.403 | 0.802 |
| | 336 | **0.588** | **0.411** | 0.614 | 0.412 | - | - | 0.594 | 0.420 | - | - | 1.415 | 0.804 |
| | 720 | **0.618** | **0.422** | 0.749 | 0.464 | - | - | 0.723 | 0.472 | - | - | 1.437 | 0.808 |
| | avg | **0.588** | 0.412 | 0.613 | **0.407** | - | - | 0.615 | 0.421 | - | - | 1.411 | 0.804 |

