# OpenReview forum: "Towards a General Time Series Forecasting Model with Unified Representation and Adaptive Transfer"
_ICML.cc/2025/Conference — ICML 2025 poster_

### Official Review · Reviewer_xtuG · 2025-03-07

**Overall Recommendation:** 4

**Summary:**

This paper introduces ReadyTS, a general time series forecasting model that learns a unified representation during pretraining and can be adaptively transferred to downstream tasks. The model employs frequency-based masking for pretraining, where specific frequency components are masked using random thresholds and flags. Additionally, a Time Series Register is trained to assign domain-specific prefixes to input samples, enhancing adaptive transferability. After fine-tuning, the proposed method demonstrates strong performance on standard time series forecasting benchmarks.

**Claims And Evidence:**

The claims are supported by clear and convincing evidence.

**Essential References Not Discussed:**

The references and related work are sufficiently cited and discussed.

**Experimental Designs Or Analyses:**

The experimental designs are generally fair and valid. The main experiments are conducted on standard LSF datasets. Notably, all methods use the same context length, which is commendable as it ensures a fair comparison.

One issue lies in the zero-shot experiment, as summarized in Table 2. The results show that **Moment** performs significantly worse in zero-shot forecasting. However, Moment is not designed for zero-shot forecasting—its original paper requires linear probing to evaluate forecasting performance. This raises concerns about its application in this experiment, making its usage unclear and its results potentially misleading. Further clarification is needed on this Moment experiment is conducted in the zero-shot setting.

**Methods And Evaluation Criteria:**

Proposed methods and evaluation criteria make sense for the problem.

**Other Comments Or Suggestions:**

It is not entirely accurate to state that the length of the frequency domain representation after rFFT is L/2+1. Rounding should be included in the case that L is an odd number.

**Other Strengths And Weaknesses:**

Strengths:
1. The proposed method addresses both pretraining and adaptation, an important yet often overlooked aspect in TSFM research.
2. The paper is well-written with a clear and logical structure.
3. The experiments are fairly conducted and include a comprehensive analysis.
4. The figures effectively illustrate the methodology, enhancing clarity and understanding.

Weakness:
1. Some descriptions lack clarity and details, making it difficult to fully understand certain aspects.

**Questions For Authors:**

1. How is the zero-shot experiment of Moment conducted? Please refer to Experimental Designs or Analyses for details.
2. Why does the efficiency analysis consider training time for full-shot models while comparing them to zero-shot models? This seems inherently unfair, as full-shot methods will naturally take more time due to the additional training process. Could you clarify the rationale behind this comparison?
3. Following the previous question, what is the efficiency of the fine-tuned ReadyTS in the full-shot setting? How does its computational cost compare to other full-shot models?
4. What is the definition of "averaged time" in Table 8? How is it computed, and is it measured per batch or across the entire dataset?
5. It is unclear how the "Aggregator" in Equation 4 is defined. Does it simply compute the mean along the dimension of $K_f$?
6. When passing the representation to the linear head, do you flatten it or compute the mean over certain dimensions?
7. Is the model sensitive to context length on downstream task? Since the model is pretrained and fine-tuned with a fixed context length of 512, it is important to examine whether it can still achieve strong performance when the downstream task involves a different context length.

**Relation To Broader Scientific Literature:**

This paper not only focuses on the pretraining of a new time series foundation model but also addresses its fine-tuning and adaptation to downstream tasks. In this field, most existing works merely focus on how to pretrain universal TSFMs and demonstrate their zero-shot performance, while largely overlooking the strategy of finetuning these models on downstream tasks. Although naive fine-tuning has been shown to improve performance, the effective adaptation of pretrained time series forecasting models remains an underexplored direction. This paper bridges that gap by exploring to use domain-specific prefix in downstream task.

**Theoretical Claims:**

No proofs for theoretical claims are included in this submission.

---

> ### Author Rebuttal · Authors · 2025-04-01
>
> We would like to sincerely thank Reviewer **xtuG** for acknowledging our presentation quality and empirical contributions, as well as the helpful comments. We will revise our paper accordingly.
>
> **Q1: How is the zero-shot experiment of Moment conducted?**
>
> **A1:**
> - Moment[1] mentioned in the paper that it can be used in a zero-shot setting by retaining its reconstruction head.
> - **Experiment details:** The zero-shot experiment of Moment is designed based on the reconstruction task in the pre-training phase. Specifically, to ensure consistency between pre-training (reconstruction) and downstream prediction tasks, we actively mask the time periods to be predicted in the input sequence, and directly use the model's reconstruction output for this part as the prediction value. We will add these descriptions to the revised paper.
>
> **Q2: Why does the efficiency analysis consider training time for full-shot models while comparing them to zero-shot models?**
>
> **A2:** We include the training time of full-shot models in the efficiency analysis mainly for **practical application considerations.** When handling a specific downstream task, actual users can choose full-shot models which need to be trained from scratch for optimal performance, or zero-shot models that can predict directly without extra training. Comparing the end-to-end efficiency of both is more meaningful for users.
>
> **Q3: What is the efficiency of the fine-tuned ReadyTS in the full-shot setting? How does its computational cost compare to other full-shot models?**
>
> **A3:**  As suggested, we have supplemented the efficiency comparison between ReadyTS and other full-shot models. As shown in the table below, ReadyTS has competitive efficiency compared with full-shot models. This is because, due to the pre-training, ReadyTS requires fewer epochs to achieve optimal performance.
>
> |Models|Parameters|Averaged time|
> |---|---|---|
> |Itransformer|3.8M|34.18s|
> |PatchTST|3.2M|35.47s|
> |TimesNet|1.8M|146s|
> |Dlinear|0.018M|24.06s|
> |ReadyTS|7.4M|28.32s|
>
> **Q4: What is the definition of "averaged time" in Table 8?**
>
> A4:  For the foundation models,  "averaged time" is the inference time averaged over running model on the entire test set, and for the specific models, we also need to add the time it takes to train on the training set  (since they can't make predictions on a test set directly).
>
> **Q5: It is unclear how the "Aggregator" in Equation 4 is defined.**
>
> A5:  The aggregator is the averaging operation along the dimension of $K_f$. For example, the outputs of the encoder have the shape [ $K_f$, batch_size, n_vars, d_model, num_patch], we aggregate them to the shape [1, batch_size, n_vars, d_model, num_patch].
>
> **Q6: When passing the representation to the linear head, do you flatten it or compute the mean over certain dimensions?**
>
> A6:  For the prediction head, we flatten the representation as previous classic works [2] before inputting it for prediction. For example, the representation of the shape [batchsize, n_vars, d_model, num_patch] is flattened out as the shape [batchsize, n_vars, d_model*num_patch].
>
> **Q7: Sensitivity of context length on downstream task. Whether it can still achieve strong performance when the downstream task involves a different context length ?**
>
> A7: **ReadyTS is not sensitive to the context length on downstream tasks.** To demonstrate this, we use 96 as the downstream input length that is significantly different from 512 as used in pre-training. The following table shows the results of ReadyTS after fine-tuning with a look-back window of 96.  Despite shorter input lengths, ReadyTS still achieves state-of-the-art performance, demonstrating effective transfer of pre-trained knowledge.
>
> |Model|ReadyTS|iTransformer|PatchTST|TimesNet|Dlinear|GPT4TS|S2IPLLM|
> |---|---|---|---|---|---|---|---|
> |Metric|MSE/MAE|MSE/MAE|MSE/MAE|MSE/MAE|MSE/MAE|MSE/MAE|MSE/MAE|
> |ETTm1|0.389/**0.389**|0.407/0.410|**0.387**/0.400|0.400/0.406|0.403/0.407|0.389/0.397|0.390/0.399|
> |ETTm2|**0.272/0.321**|0.288/0.332|0.281/0.326|0.291/0.333|0.350/0.401|0.285/0.331|0.278/0.327|
> |ETTh1|**0.432/0.426**|0.454/0.447|0.469/0.454|0.458/0.450|0.456/0.452|0.447/0.436|0.444/0.431|
> |ETTh2|**0.376/0.393**|0.383/0.407|0.387/0.407|0.414/0.427|0.559/0.515|0.381/0.408|0.378/0.402|
> |Weather|**0.257/0.276**|0.258/0.278|0.259/0.281|0.259/0.287|0.265/0.317|0.264/0.284|0.266/0.284|
> |Electricity|**0.176/0.268**|0.178/0.270|0.205/0.290|0.192/0.296|0.354/0.414|0.205/0.290|0.195/0.285|
> |Traffic|0.440/**0.276**|**0.428**/0.282|0.481/0.304|0.620/0.336|0.625/0.383|0.488/0.317|0.467/0.305|
>
> [1] Goswami, M., Szafer, K., Choudhry, A., Cai, Y., Li, S., & Dubrawski, A. (2024). Moment: A family of open time-series foundation models. *arXiv preprint arXiv:2402.03885*.
>
> [2] Nie, Y., Nguyen, N. H., Sinthong, P., & Kalagnanam, J. (2022). A time series is worth 64 words: Long-term forecasting with transformers. *arXiv preprint arXiv:2211.14730*.

---

> > ### Comment · Reviewer_xtuG · 2025-04-02
> >
> > Thank you for the detailed reply. All points and concerns have been properly addressed and clearly explained.
> >
> > However, I remain concerned about the use of Moment for zero-shot forecasting. While Moment is suitable for zero-shot representation learning or anomaly detection, it is not inherently designed for zero-shot forecasting. The authors’ implementation is theoretically acceptable, but the model itself is not a natural fit for this task. In fact, the original paper used Moment-LP for evaluation, and your experimental results show that using the reconstruction head for forecasting performs much poorly.
> >
> > If the authors wish to retain these results, please provide a clear explanation of how zero-shot forecasting is conducted using Moment, and explicitly state that Moment does not officially support zero-shot forecasting in this manner.
> >
> > The remaining issues are clear to me. After the consideration, I will keep my score.

---

> > > ### Author Response · Authors · 2025-04-02
> > >
> > > Thank you for your positive reception of our new experimental results and clarifications.
> > > - Since the experiment in the Table 2 is a comparison of different time series foundation models under the zero-shot setting, we reported the zero-shot results of Moment.
> > > - We fully understand your concerns. Next, we will emphasize the following in our revised paper:
> > >     - The zero-shot experiment of Moment is designed based on the reconstruction task in the pre-training phase. Specifically, to ensure consistency between pre-training (reconstruction) and downstream prediction tasks, we actively mask the time periods to be predicted in the input sequence, and directly use the model's reconstruction output for this part as the prediction value.
> > >     - Moment itself is not designed for zero-shot prediction and does not officially support zero-shot forecasting in this manner.
> > > - In addition, we will supplement the the following results of Moment_lp in Table 1.
> > >
> > > | Datasets | ETTh1 | ETTh2 | ETTm1 | ETTm2 | Weather | Electricity | Traffic |
> > > | --- | --- | --- | --- | --- | --- | --- | --- |
> > > | Metrics | MSE / MAE | MSE / MAE | MSE / MAE | MSE / MAE | MSE / MAE | MSE / MAE | MSE / MAE |
> > > | Moment_lp | 0.418 / 0.436 | 0.352 / 0.395 | 0.344 / 0.379 | 0.258 / 0.318 | 0.228 / 0.270 | 0.165 / 0.260 | 0.415 / 0.293 |

---

### Official Review · Reviewer_3YLu · 2025-03-14

**Overall Recommendation:** 2

**Summary:**

The paper proposes a pre-trained time series model for forecasting.
The model distinguishes itself from other pre-trained models by three main aspects: (1)  frequency-based masking in pre-training, (2) the time series register, (3) and a double objective of forecasting and reconstruction in pre-training.
An empirical study shows the few-shot and zero-shot performance of the model.

**Claims And Evidence:**

While the authors aim to support the main claims with according experiments, including ablations, I have some concerns regarding the current evaluation setup (see Experimental Designs Or Analyses)

Apart from the major claims, the paper sometimes tends to state hypotheses as facts --- or sentences came out of the middle without any proper connections to the previous text.  to name a few examples in:
- "Thus, they only transfer the same generalized representation to different target domains, called direct transfer, which limits the model’s effectiveness in specific downstream tasks." (line 81)  -> not yet clear whether this is the case, general representation in e.g. LLM work well.
- "By decomposed frequency learning, we can obtain the general representations" (line 239) -> This section start is out of the blue. While I understand that the authors aim to do that with decomposed frequency learning, it is unclear if you can do it at that point in the paper.

Hence, a little bit of resharpening of the text could improve the paper quite  a bit

**Essential References Not Discussed:**

I am not aware of any essential references that are missing.

**Experimental Designs Or Analyses:**

I have concerns regarding the validity and generalizability of the evaluation. My concerns fall into two main areas:

### 1 - The generalizability of the evaluation benchmark in general
The method is evaluated on 4 datasets (ETT is reported as four separate datasets). Although this dataset is frequently used in related literature, there are reasonable concerns regarding the generalizability and validity of these. See for example [1,2].
Especially for a pre-trained time series model, which should generalize across a wide spectrum of time series and where evaluation on further evaluation datasets is straightforward and comes with a limited amount of work, as the model does not require individual training in a zero-shot mode, I think a more comprehensive evaluation should be considered.

Hence, I would strongly suggest making use of recent advances in forecast benchmarking, e.g., using the givt-eval benchmark [3] or the benchmarks from [4], which provide more comprehensive insights - especially also because they allow a comparison beyond self-trained baseline methods. Overlaps of individual datasets with the pre-training corpus should be highlighted by the authors (An aggregation excluding these should be straightforward)

### 2 - Selection of pre-trained models in comparision
For Chronos, TimesFM, and Morai, new model versions are published. (Chronos Bolt, TimesFM 500M, Moirai-1.1)
Additionally, for Chronos and MOIRAI there are multiple sizes available also for version that is used by the authors . For example, for Chronos the authors used "Chronos Small" without any justification, although "Chronos Base" typically shows improved performance.
While I understand that the former might be because they are concurrent work, the latter should at least be noted in the paper, and the full results of all models should be provided in the appendix


[1] Brigato, Lorenzo, et al. "Position: There are no Champions in Long-Term Time Series Forecasting." arXiv preprint arXiv:2502.14045 (2025).
[2] Invited Talk by Christoph Bergmeir - Fundamental limitations of foundational forecasting models: The need for multimodality and rigorous evaluation, Time Series in the Age of Large Models Workshop NeurIPS 2024
[3] Aksu, Taha, et al. "GIFT-Eval: A Benchmark For General Time Series Forecasting Model Evaluation." arXiv preprint arXiv:2410.10393 (2024).
[4] Ansari, Abdul Fatir, et al. "Chronos: Learning the Language of Time Series." Transactions on Machine Learning Research. 2024.

**Methods And Evaluation Criteria:**

In general, the approach to empirically validate the model (components) by comparing to different baselines models and ablations with the given metrics make sense.
However, I have some concerns regarding the evaluation setup in general (see Experimental Designs Or Analyses)

**Other Comments Or Suggestions:**

- Table2 - Traffic: ReadyTS is marked as best model, however its worse than Chronos
- Appendix A.10.4. ZERO-SHOT RESULT: There is placeholder text

**Other Strengths And Weaknesses:**

Strengths:
- The paper included ablations analysis of different components.
- The paper is nicely structured and relatively easy to follow (however, parts are unclear to me - see questions below)


Weakness:
- Limited evaluation benchmark for a pre-trained time series model
- The multi-loss setting might be difficult to balance

**Questions For Authors:**

- Do the individual masked sequences of one time series get processed independently by the transformer encoder?
- How do you handle different lengths of time series when the full time series is projected into the embedding $x_e$, which is used in TS-Register.
- In pre-training, the authors state that "gradients of the prediction heads are skipped at back-propagation". Does that mean that the prediction loss effectively only trains the prediction head, as no gradient flows backward through the rest of the model?
- The authors state that "the parameters of the reconstruction decoder are copied to the prediction decoder during forward propagation". Is this not the same as simply using the one decoder for both things? Why do we distinguish between reconstruction and prediction decoder if this is the case?

**Relation To Broader Scientific Literature:**

The method is located in the field of pre-trained time series models.
The two key contributions (frequency-based masking and time series register) are to the best of my knowledge, novel in the field of pre-trained models and might improve the field overall.

**Theoretical Claims:**

There are no theoretical claims or proofs in the paper.

---

> ### Author Rebuttal · Authors · 2025-04-01
>
> We would like to sincerely thank Reviewer **3YLu** for providing detailed review and insightful comments regarding the model design and empirical study. We will revise our paper accordingly.
>
> **Q1: Resharpening of the text**
>
> **A1:** Thank you for your valuable suggestions on the presentation of the article.
>
> - We would like to clarify that the generalized representations refer to those learned by time series pre-training frameworks, not LLMs. Thus, we update the description as follows. "**However, as shown in Figure 1(a), existing time series pre-training frameworks focus mainly on learning generalized time series representation during pre-training and overlook domain-specific representation. Such direct transferring the generalized time series representation to different target domains limits the effectiveness of these frameworks in specific downstream tasks.**"
> - We would like to clarify that since we have discussed the intuition of learning generalized time series representation in Section 3.2 Decomposed frequency learning, we simply state "By decomposed frequency learning, we can obtain the general representations" at the beginning of Section 3.3. To further clarify the description, we thus update the sentence as "**As described in Sec. 3.2, we can obtain the general representations by decomposed frequency learning.**"
>
> **Q2: Limited evaluation benchmark for a pre-trained time series model.**
>
> **A2:** We add a more comprehensive fev benchmark you mentioned. The fev benchmark focuses on the model's short-term prediction ability, while we design ReadyTS that shows more expertise at long-term predictions of 96-720 steps. Nevertheless, ReadyTS still exhibited competitive performance and superior efficiency compared to other foundation models.
>
> |Model|Average Relative Error|Median inferencetime(s)|Training corpus overlap(%)|
> |---|---|---|---|
> |Moirai-large|**0.791**|14.254|81.5%|
> |TimesFM|0.882|0.396|14.8%|
> |Chronos-large|0.824|26.827|0%|
> |ReadyTS|0.833|0.334|31%|
> |seasonal_naive|1|**0.096**|0%|
>
> **Q3:  Selection of pre-trained models in comparision.**
>
> A3:
>
> - We select five foundation models for comparison in zero-shot setting,  including Timer-67M, MOIRAI-large, TimesFM-200M, Chronos-small and Moment-base.
> - We appreciate your understanding that these are concurrent work, and we will provide the full results of all models in the revised paper.
> - Due to the low inference efficiency of Chronos-large, we chose the small version for comparison. Subsequently, we add Chronos-large as a baseline in the fev benchmark.
>
> **Q4: The multi-loss setting might be difficult to balance.**
>
> A4: Due to space limitations, we respectfully refer you to check A2 in Reviewer EPkr.
>
> **Q5: Textual errors in the paper.**
>
> A5: The result of Chronos in Table2 - Traffic should be 0.615, and ReadyTS achieves the best result. We will correct this in the revised paper.  We will also remove the placeholder text from Appendix A.10.4 and add explanatory text to A.10.1, 10.2, and 10.4 to correspond to the tables.
>
> **Q6: Do the individual masked sequences of one time series get processed independently by the transformer encoder?**
>
> A6:  Yes.
>
> **Q7: How to handle different lengths of time series when the full time series is projected into the embedding $x_e$, which is used in TS-Register?**
>
> A7:
>
> - **The register can handle input time series with different lengths without failing.** To adapt to different input lengths, we create a new linear projection layer for the new input length and update its parameters during fine-tuning.
> - **Experiment:** We validate the effectiveness of the register under an input length of 96 which is different from 512. **The average results of all predicted lengths in the table below demonstrate that the effectiveness of the register is maintained.**
>
> ||w register|w/o register|
> |---|---|---|
> |Metric|MSE/MAE|MSE/MAE|
> |ETTm1|**0.389/0.389**|0.392/0.390|
> |ETTm2|**0.272/0.321**|0.287/0.323|
> |ETTh1|**0.432/0.426**|0.436/0.431|
> |ETTh2|**0.376/0.393**|0.381/0.398|
> |Traffic|**0.440**/**0.276**|0.450/0.288|
> |Weather|**0.257/0.276**|0.264/0.275|
> |Solar|**0.230/0.255**|0.249/0.270|
> |Electricity|**0.182**/**0.268**|0.189/0.275|
>
>
> **Q8:The gradients of the prediction heads.**
>
> A8: As shown in Figure 2, the gradients of the prediction loss only skip the decoder part and still affect the rest of the model.
>
> **Q9:The reason of distinguish between reconstruction and prediction decoder.**
>
> A9:  In forward-propagation, copying the parameters of the reconstruction decoder to the prediction decoder is same as using one decoder for both tasks. However, in back-propagation, the gradient of the prediction loss skip the decoder, while the reconstruction loss is propagated normally. The reason we designed two decoders is that we do two tasks in pre-training. However, considering the convenience and effectiveness of training, we share the parameters of the two decoders and let the gradients of the prediction loss skip the decoder.

---

> > ### Comment · Reviewer_3YLu · 2025-04-02
> >
> > Thank you for the response. I have some follow-up questions:
> >
> > A1: My concern was not to clarify weather the time series model or LLM general representation but how you conclude generalized representation "limits the effectiveness of these frameworks". Could you elaborate on how you arrived at this conclusion? Which downstream task are you referring to exactly, and what is the empirical evidence for that? I want to highlight your response of A2, which shows that general representation (Chronos, Chronos-Bolt, TimesFM) seem to provide often better results than ReadyTS on the fev benchmark.
> >
> > A2: Thank you for providing the fev results and clarifying that ReadyTS mostly aims at long-term predictions, if I correctly understand the authors? I would suggest discussing this also in the paper.
> >
> > A7: How do you handle this in a zero-shot setting?
> >
> > A8/9: How is that implemented practically? By copying and freezing (for the prediction decoder) the parameter in each forward pass during training?

---

> > > ### Author Response · Authors · 2025-04-03
> > >
> > > Thank you for your thoughtful questions and valuable feedback. We appreciate the opportunity to clarify and refine our responses.
> > >
> > >
> > > **A1:**
> > > - Thank you for your insightful question. To clarify, our intention is not to suggest that general representations are ineffective. Instead, we aim to highlight that incorporating domain-specific representations can further improve the performance of downstream tasks. This point is supported by the t-SNE visualization in Figure 1(b) and the ablation studies in Table 3 (TS-Register), which demonstrate the value of domain-specific representations.
> > > - To better articulate this, we have revised the description as follows: "However, as shown in Figure 1(a), existing time series pre-training frameworks primarily focus on learning generalized time series representations during pre-training while overlooking domain-specific representations. **While generalized representations are essential, directly transferring them to specific downstream tasks without incorporating domain-specific information leaves room for improvement**“
> > >
> > > **A2:**
> > > - Thank you very much for your suggestion. We will supplement this in the revised paper.
> > >
> > > **A7:**
> > > - In the zero-shot setting, when the length of the input sequence differs from 512, we upsample or downsample the input sequence to match the length of the pretrained input, enabling zero-shot predictions.
> > >
> > > **A8/A9:**
> > > - Yes, you are correct. During the forward pass, the parameters of the reconstruction decoder are copied to the prediction decoder and frozen. During the backward pass, the gradients of the prediction loss skip the prediction decoder and directly propagate back to the backbone.

---

### Official Review · Reviewer_DsF3 · 2025-03-14

**Overall Recommendation:** 4

**Summary:**

The work introduces a new method of learning foundational time-series models from pre-training on heterogenous datasets via decomposed frequency learning. The key idea is to extract multiple frequency representation via FFT and using masking in frequency domain as well when reconstructing the time-series. They also have domain-specific "register tokens" for better domain-specific generaliziation for different down stream tasks.

**Claims And Evidence:**

The main problem discussed was pre-traning with hetergenous multi-domain datasets. This was addressed in their method.

**Essential References Not Discussed:**

Recent baselines like LPTM, TIme-MOE, Chronos-bolt are not included.

**Experimental Designs Or Analyses:**

Yes. The fine-tuned, few-shot and zero-shot experiments makes sense. The datasets and baselines are valid.

**Methods And Evaluation Criteria:**

Yes. The methodology and evaluation look valid and easy to understand.

**Other Comments Or Suggestions:**

NA

**Other Strengths And Weaknesses:**

Weaknesses
1. Lack of some baselines
2. Runtime complexity and pre-train compute resources can be mentioned.

**Questions For Authors:**

See weaknesses above.

**Relation To Broader Scientific Literature:**

This method seems to be a significant contribution in the line of recent foundational time-series models.

**Theoretical Claims:**

NA

---

> ### Author Rebuttal · Authors · 2025-04-01
>
> We would like to sincerely thank Reviewer **DsF3** for providing a detailed review and insightful comments. We will revise our paper accordingly.
>
> **Q1: Lack of some baselines**
>
> A1: Based on your suggestions, we add some recent baselines: LPTM, Time-MOE-large and Chronos-bolt-base. As shown in the table below, ReadyTS also exhibited competitive performance compared to these baselines.
>
> | Model | ReadyTS | LPTM | Time-MOE | Chronos-bolt |
> | --- | --- | --- | --- | --- |
> | Metric | MSE  | MSE  | MSE  | MSE  |
> | ETTm1 | 0.525 | 0.493 | 0.484 | **0.475** |
> | ETTm2 | **0.299** | 0.378 | 0.512 | 0.508 |
> | ETTh1 | **0.401** | 0.414 | 0.436 | 0.447 |
> | ETTh2 | **0.346** | 0.398 | 0.479 | 0.366 |
> | Weather | **0.265** | 0.274 | 0.319 | 0.268 |
>
> **Q2:  Runtime complexity and pre-train compute resources can be mentioned.**
>
> A2:
>
> - **Runtime complexity :** In Section 4.3, we compare the efficiency of the foundation model by computing the inference time. Additionally, the following table shows the memory usage for different foundation models (using the ETTh2 dataset as an example with a batch size of 1).
>
> | Model | ReadyTS | Timer | Moirai | Chronos | TimesFM | Moment |
> | --- | --- | --- | --- | --- | --- | --- |
> | Menmory memory usage (MB) | **577** | 1435 | 2009 | 10269 | 1395 | 4486 |
> - **Pre-train compute resources**: In the pre-training stage, we used 2 NVIDIA A800 80GB GPUs for only about 20 hours of training.

---

### Official Review · Reviewer_EPkr · 2025-03-15

**Overall Recommendation:** 4

**Summary:**

This paper builds a foundation model ReadyTS from two aspects:unified representations from heterogeneous multidomain time series data;domain-specific features to enable adaptive transfer across various downstream scenarios. First, this paper leverages frequency-based masking and reconstruction to decompose coupled semantic information in time series. Second, this paper proposes Time Series Register, which captures domain-specific representations during pre-training and enhances adaptive transferability to downstream tasks. The model achieves the SOTA performance in time series forecasting, as well as the few-shot and zero-shot scenarios

**Claims And Evidence:**

Yes.

**Essential References Not Discussed:**

There are no essential references not discussed

**Experimental Designs Or Analyses:**

checked

**Methods And Evaluation Criteria:**

Yes. The proposed methods and the evaluation criteria make sense.

**Other Comments Or Suggestions:**

n/a

**Other Strengths And Weaknesses:**

Strengths:

1	This paper is well-written and easy to follow.

2	The proposed method is with clear motivations and reasonable technical designs. The designs of decomposed frequency learning and time series register are also interesting.

3	SOTA performance and it shows efficiency advantages compared with existing foundation models.

4	The evaluation settings and model analysis are extensive.

Weaknesses:

1	Some designs of the model pre-training need to be clarified. How is the register initialized? Is it trained together with the foundation model? Why is the prediction task needed during the pre-training?

2	In Equation (12), are there any trade-off hyper-parameters needed between these different losses to balance their effects? It may need some experiments regarding this.

3	Figure 1(b) needs more explanations. How are the hidden representations extracted in direct transfer and adaptive transfer respectively?

**Questions For Authors:**

refer to the weaknesses

**Relation To Broader Scientific Literature:**

I think the foundation model of time series forecasting is an interesting topic

**Theoretical Claims:**

It seems this paper does not have theoretical claims or proofs.

---

> ### Author Rebuttal · Authors · 2025-04-01
>
> We would like to sincerely thank Reviewer **EPkr** for acknowledging our technical novelty and effectiveness, as well as the insightful comments. We will revise our paper accordingly.
>
> **Q1:  How is the register initialized? Is it trained together with the foundation model? Why is the prediction task needed during the pre-training?**
>
> **A1:**
>
> - **Register initialization:**  the vectors in register are randomly initialized from a standard normal distribution  $\mathcal{N}(0,1)$.
> - The register is trained together with the foundation model, while the gradient of register loss does not affect the backbone.
> - To further enhance the effectiveness of this architecture for time series prediction tasks and enable few-shot and zero-shot capabilities, aiming to build a general time series forecasting model, we add prediction heads to improve prediction accuracy.
>
> **Q2:  In Equation (12), are there any trade-off hyper-parameters needed between these different losses to balance their effects? It may need some experiments regarding this.**
>
> **A2:**
>
> - We did not use a hyper-parameter to balance the loss functions because our experiments found that the model is not sensitive to the weights of the multi-losses. In the experiment, we introduce a hyper-parameter $\lambda$ , and define $\mathcal{L} _{\text{pretrain}} = \lambda \mathcal{L} _{\text{reconstruction}} + (1 - \lambda) \mathcal{L} _{\text{prediction}} + \mathcal{L} _{\text{register}}$. Since the register loss only constrains the parameter updates of the register, its gradient does not influence the backbone of the model. Therefore, the register loss does not cause an imbalance in the training of the model.
> - We vary $\lambda$'s value and report results in the table below.  **As ReadyTS is not sensitive to changes of $\lambda$, balancing the loss of the model is not challenging. Therefore, our final loss function does not contain $\lambda$.**
>
> | lambda | 0.2 | 0.4 | 0.6 | 0.8 | Standard Deviation |
> | --- | --- | --- | --- | --- | --- |
> |  Metrics| MSE / MAE | MSE / MAE | MSE / MAE | MSE / MAE | MSE / MAE |
> | ETTh1 | 0.3973 / 0.4199 | 0.3978 / 0.4193 | 0.3978 / 0.4205 | 0.3996 / 0.4230 | 0.0008 / 0.0014 |
> | ETTh2 | 0.3339 / 0.3790 | 0.3347 / 0.3802 | 0.3369 / 0.3822 | 0.3351 / 0.3830 | 0.0011 / 0.0016 |
> | ETTm1 | 0.3500 / 0.3733 | 0.3512 / 0.3747 | 0.3492 / 0.3717 | 0.3479 / 0.3719 | 0.0012 / 0.0012 |
> | ETTm2 | 0.2538 / 0.3111 | 0.2534 / 0.3095 | 0.2512 / 0.3092 | 0.2505 / 0.3092 | 0.0014 / 0.0007 |
>
> **Q3:  Figure 1(b) needs more explanations. How are the hidden representations extracted in direct transfer and adaptive transfer respectively?**
>
> **A3:**  We select three datasets (Pems08, PSRA, Electricity) from transport, nature and energy domains respectively and compare the differences in hidden representations between direct transfer and adaptive transfer. Specifically, direct transfer refers to the case where domain specific information is not considered, while adaptive transfer considers domain specific information that is learned by register tokens. We visualized the output of the encoder's hidden representations using t-SNE. The description of the setting would be updated in the revised paper.

---

### Decision · Program_Chairs · 2025-05-01

**Decision:**

Accept (poster)

**Comment:**

This paper introduces ReadyTS, a general time series forecasting model that learns a unified representation during pretraining and can be adaptively transferred to downstream tasks. The model employs frequency-based masking for pretraining, where specific frequency components are masked using random thresholds and flags. Additionally, a Time Series Register is trained to assign domain-specific prefixes to input samples, enhancing adaptive transferability. After fine-tuning, the proposed method demonstrates strong performance on standard time series forecasting benchmarks.


The strengths are: 1) The proposed method is with clear motivations and reasonable technical designs and designs of decomposed frequency learning and time series register are also interesting; 2) The proposed method addresses both pretraining and adaptation, an important yet often overlooked aspect in TSFM research ; 3) The evaluation settings are extensive, and it shows efficiency advantages compared with existing foundation models; 4) This paper is well-written and easy to follow. One major concern raised in the initial reviews is limited evaluation benchmark for a pre-trained time series model. In the rebuttal, additional experiments were added to address this concern.